# Control of coordinatively unsaturated Zr sites in ZrO$_2$ for efficient C–H bond activation

Yaoyuan Zhang[1,2], Yun Zhao [1], Tatiana Otroshchenko[1], Henrik Lund [1], Marga-Martina Pohl[1], Uwe Rodemerck[1], David Linke [1], Haijun Jiao[1], Guiyuan Jiang[2] & Evgenii V. Kondratenko [1]

Due to the complexity of heterogeneous catalysts, identification of active sites and the ways for their experimental design are not inherently straightforward but important for tailored catalyst preparation. The present study reveals the active sites for efficient C–H bond activation in C$_1$–C$_4$ alkanes over ZrO$_2$ free of any metals or metal oxides usually catalysing this reaction. Quantum chemical calculations suggest that two Zr cations located at an oxygen vacancy are responsible for the homolytic C–H bond dissociation. This pathway differs from that reported for other metal oxides used for alkane activation, where metal cation and neighbouring lattice oxygen form the active site. The concentration of anion vacancies in ZrO$_2$ can be controlled through adjusting the crystallite size. Accordingly designed ZrO$_2$ shows industrially relevant activity and durability in non-oxidative propane dehydrogenation and performs superior to state-of-the-art catalysts possessing Pt, CrO$_x$, GaO$_x$ or VO$_x$ species.

[1] Leibniz-Institut für Katalyse e.V. an der Universität Rostock, Albert-Einstein-Straße 29a, Rostock 18059, Germany. [2] State Key Laboratory of Heavy Oil Processing, China University of Petroleum Beijing, Beijing 102249, China. These authors contributed equally: Yaoyuan Zhang, Yun Zhao. Correspondence and requests for materials should be addressed to G.J. (email: jianggy@cup.edu.cn) or to E.V.K. (email: evgenii.kondratenko@catalysis.de)

The ability of scientists to elaborate and apply fundamental principles for rational design of heterogeneous catalysts is decisive for the global sustainable development and the environment protection applications. One possible way in this direction is to identify the active sites or key atomic structures[1,2] and, for the most part, to use this knowledge for catalyst design and preparation. For example, owing to the well-defined structure (size and/or shape) and composition of nanoparticles of metals or metal oxides, they are successfully used for the development of supported catalysts with controlled properties[3–8]. Modifying structure of bulk metal oxides at a nanoscale level also has striking effects on their physico-chemical and catalytic properties[9–13]. The present study elucidates at molecular level the effects of nanostructure of bare $ZrO_2$ on the activity and selectivity in the non-oxidative dehydrogenation of ethane, propane or iso-butane to the corresponding olefins.

In comparison with oil-based cracking technologies, which provide a major part of $C_2$–$C_4$ olefins for the chemical industry, ethane steam cracking and the dehydrogenation of propane or iso-butane are more environmentally friendly processes because they are based on natural/shale gas containing significantly less impurities than the oil-based feedstock. In addition, the dehydrogenation of propane (PDH) has been developed to close the gap between the demand and supply of propene[14,15]. This olefin is second to ethene as an important building block of the chemical industry with an annual production of about 80 million metric tons[15]. Contribution of PDH technology to the overall propene production will grow because new plants are under construction[16]. There are, however, flaws related to the commercial catalysts with supported $CrO_x$ or Pt species[14,15]. According to the U.S. Occupational Safety and Health Administration[17], workplace exposure to Cr(VI) may cause various health effects. A key drawback of Pt-based catalysts is their cost and deactivation triggered by sintering of Pt species. To re-disperse platinum, the catalysts are treated by ecologically harmful $Cl_2$ or Cl-containing compounds[15]. Recently, we developed eco-friendly and cost-efficient catalysts on the basis of $ZrO_2$, which had, however, to be promoted with metal oxides and contain supported Ru or Rh NP to show high activity in the non-oxidative dehydrogenation of propane, n-butane or iso-butane[18–20]. Coordinatively unsaturated Zr ($Zr_{cus}$) and neighbouring lattice oxygen were suggested to participate in the dehydrogenation reaction. Yet, the promoter used to purposefully create $Zr_{cus}$ sites either increased or decreased the activity hence proving the limitations of this classical approach for tailored catalyst design.

Here, we describe how nanostructuring of $ZrO_2$ crystallites enables the identification of the nature of active sites for efficient C–H bond activation and the control of their concentration without the usage of any dopant or supported species. Acid–base, redox and electrical conductivity properties of $ZrO_2$ can also be controlled through crystallite size. A structural model of the active site is established owing to our multidisciplinary approach combining density functional theory (DFT) calculations with a number of complementary experimental methods including catalytic tests. The active site consists of two Zr cations located at an oxygen vacancy, which homolytically break the C–H bond in alkanes. The kind of this site differs from that previously suggested by some of us for doped $ZrO_2$-based catalysts[18,19] and by other researchers for different metal oxides used for PDH[21–23], where metal cation and neighbouring lattice oxygen were assumed to form the active site. Bare $ZrO_2$ designed especially to maximize the concentration of the active sites shows industrially relevant performance in comparison with commercial-like catalysts containing $CrO_x$ or Pt species and other alternative state-of-the-art catalysts.

## Results

**Controlling activity and selectivity of $ZrO_2$.** The scientific background for our study was the fact that the relative ratio of coordinatively unsaturated sites on the surface of metal oxides to their regular counterparts depends on the size of crystallites[24]. To check the influence of the nanostructure of bare $ZrO_2$ for C–H bond activation in lower alkanes, we synthesized about 40 $ZrO_2$ samples (Supplementary Table 1). Powder X-ray diffraction (XRD) analysis proved that they are mainly composed of the monoclinic phase (Supplementary Fig. 1) but differed in the average size of crystallites determined from the ($\bar{1}11$) and (111) XRD reflexes. As seen in Fig. 1a, the rate of propene formation in PDH decreased by a factor of up to 70 with an increase in the size from 7 to 45 nm. There were no such strong differences between the catalysts in terms of the rate of propene formation if non-catalytic propane dehydrogenation played a significant role. The latter process is actually not relevant under the reaction conditions applied as proven by separate tests[19] without catalysts.

It is also worth mentioning that for $ZrO_2$ samples calcined at temperatures below 550 °C, the size of crystallites was determined after performing the PDH reaction (Supplementary Table 2). In addition to the crystallite size, long-range order of lattice oxygen and zirconium cations within at least a few nanometres in the crystal lattice of $ZrO_2$ should be relevant for achieving high PDH activity. This conclusion is based on the fact that XRD-amorphous $ZrO_2$ showed very low activity (Fig. 1a). The obtained relationship between the rate of propene formation and the size of crystallites is also valid when the rate is calculated with respect to catalyst surface area (Supplementary Fig. 2a). Thus, a simple effect of the crystallite size on the surface area and accordingly on the activity can be excluded, as the rate related to catalyst surface area increases with the area (Supplementary Fig. 2b-c).

No effect of the area would be visible if this catalyst property determined the activity exclusively. Moreover, although amorphous $ZrO_2$ samples ($ZrO_2\_24$ and $ZrO_2\_26$ in Supplementary Table 1) possess the highest surface area, they showed the lowest activity. The PDH activity of $ZrO_2$ is governed by the number of the active sites, which depends on the size of crystallites. A further experimental support for the importance of the latter catalyst property for PDH is the fact that apparent activation energy of propene formation depends on the size. The energy increases from about 140 to 300 kJ mol$^{-1}$ with an increase in the size from 9 to 43 nm (Supplementary Fig. 3). This effect is related to the kind of catalytically active sites as will be demonstrated below when taking further catalytic and characterization studies as well as DFT calculations into account.

The activity-size relationship in Fig. 1a can be explained by an increase in the concentration of the catalytically active $Zr_{cus}$ sites with a decrease in crystallite size. To validate the importance of such sites for propene formation, catalytic tests were performed with $ZrO_2$ treated with either $H_2$ or CO at 550 °C before testing in PDH at the same temperature. The idea was to create additional $Zr_{cus}$ sites through removal of lattice oxygen from $ZrO_2$ in form of $H_2O$ or $CO_2$. The rate of propene formation increased by a factor of 2 after 6 h on $H_2$ stream in comparison with the non-treated sample (Fig. 1c).

When CO was used as a reducing agent, a seven-time increase in the rate of propene formation was achieved after only 30 min treatment. Such distinctive effects of the reducing agents on the increase in the rate are related to their ability to remove lattice oxygen from $ZrO_2$ as proven by temperature-programmed reduction (TPR) tests with $H_2$ or CO (Supplementary Fig. 5). Our DFT calculations also predict that oxygen vacancies are easier formed through removal of lattice oxygen by CO than by $H_2$ (Supplementary Table 3). The relevance of the so-generated surface defects for propene formation is further supported by the

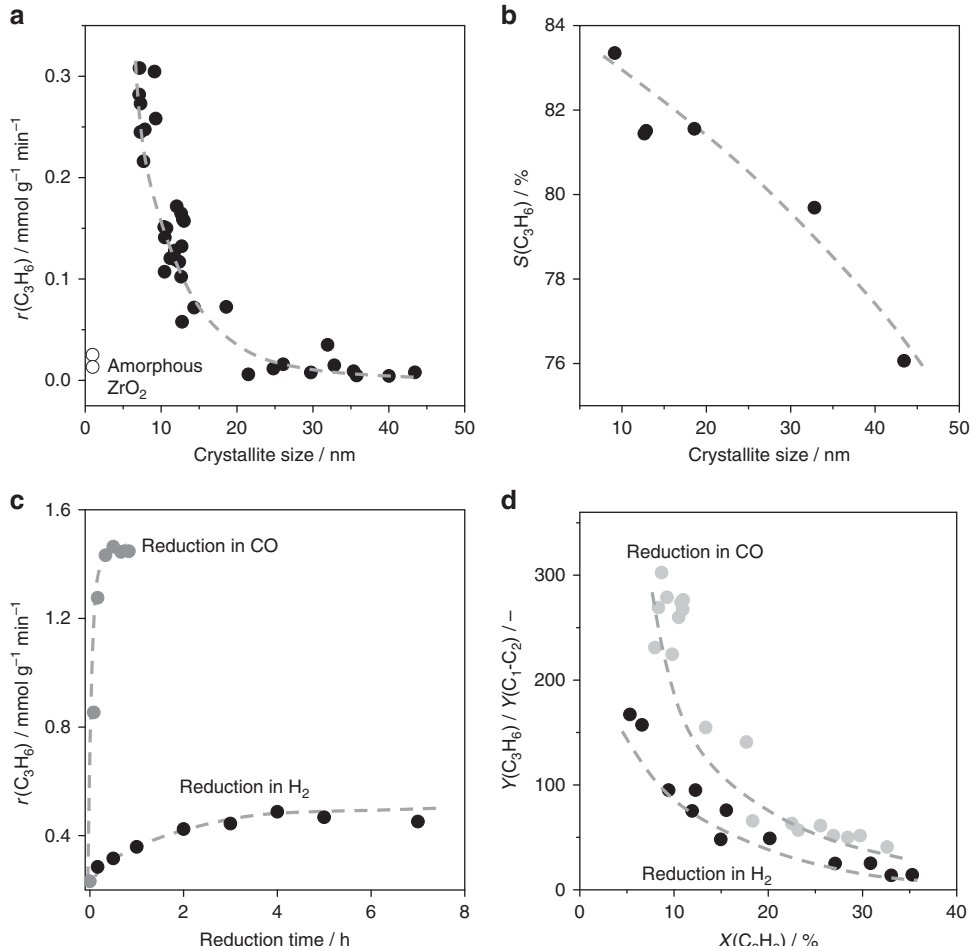

**Fig. 1** Catalytic property in PDH. **a** The rate of propene formation ($r(C_3H_6)$) versus the size of crystallites of differently prepared $ZrO_2$ (Supplementary Table 1). **b** An integral selectivity to propene ($S(C_3H_6)$) calculated from the number of $C_3H_8$ moles converted and $C_3H_6$ moles formed within 1 h on $C_3H_8$ stream (Supplementary Fig. 4 and Supplementary Equation 2) over $ZrO_2$ with different crystallite sizes. **c** $r(C_3H_6)$ over $ZrO_2$ after reduction in $H_2$ or CO. **d** The effect of catalyst treatment by CO or $H_2$ on the ratio of yields of propene to cracking products ($Y(C_3H_6)/Y(C_1-C_2)$). $ZrO_2$ with the highest $S(C_3H_6)$ in **b** was used in **c** and **d**. All these tests were performed at 550 °C using a feed containing 40 vol% $C_3H_8$ in $N_2$

higher ratio of the yield of propene ($Y(C_3H_6)$) to that of cracking products ($Y(C_1-C_2)$) in a broad range of propane conversion obtained over the CO-treated catalyst in comparison to its $H_2$-treated counterpart (Fig. 1d). This result suggests that propene and cracking products are formed on different sites.

The use of $ZrO_2$ composed of small crystallites is also beneficial for propene selectivity, which increased with a decrease in the crystallite size (Fig. 1b). Such positive effect holds over a broad range of propane conversion (Supplementary Fig. 6). An insight into the formation and the kind of carbon deposits, which are the main undesired reaction product, was derived from the operando UV-vis analysis upon PDH over three $ZrO_2$ samples with 9.1, 13.0 or 43.4 nm crystallites. The obtained UV-vis spectra expressed as the relative Kubelka–Munk function ($F(R_{rel})$ in Eq. (2)) after different times on propane stream are shown in Fig. 2a–c. $F(R_{rel})$ increased across the entire range of the UV-vis spectrum with rising time on propane stream. This increase must have occurred due to the deposition of coke species because the UV-vis spectra of $H_2$-treated catalysts differ strongly (Supplementary Fig. 7). Absorption bands with the maxima at about 300, 420, 600 and 750 nm can be tentatively identified in the spectra of $ZrO_2$ under PDH conditions. The latter two signals can be ascribed to polyaromatic graphitic species as previously suggested for PDH over a Cr-containing catalyst[25]. Temporal changes of the

intensity of these bands under PDH conditions follow the same trend thus suggesting that they belong to the same carbon-containing species (Fig. 2d–f). The bands with the maxima in UV range (at 300 and 420 nm) should originate from low-condensed aromatic species, which differ from polyaromatic graphitic species also absorbing light above 500 nm. Different carbon-containing species are formed with different rates as seen in Fig. 2d–f showing temporal changes in the intensity of the corresponding bands with rising time of propane stream. When analysing the UV-vis spectra in Fig. 2a–c, it becomes obvious that the relative ratio of the intensity of absorption bands at 300 and 420 nm to that of the bands at higher wavelengths depends on the size of $ZrO_2$ crystallites. The larger the size, the higher the fraction of polyaromatic graphitic species (absorption bands above 500 nm) is expected. In addition, there is an induction period before the UV-vis spectra start to change under the PDH conditions (Supplementary Fig. 8). Such delay may indicate that some coke precursors (seeds) must be initially formed before coke formation can proceed. Importantly, the duration of this induction period increased with a decrease in the size of $ZrO_2$ crystallites (Supplementary Fig. 8).

On the basis of the above discussion, the effect of the size of $ZrO_2$ crystallites on coke formation can be explained as follows. From a mechanistic point of view, adsorbed propene molecules

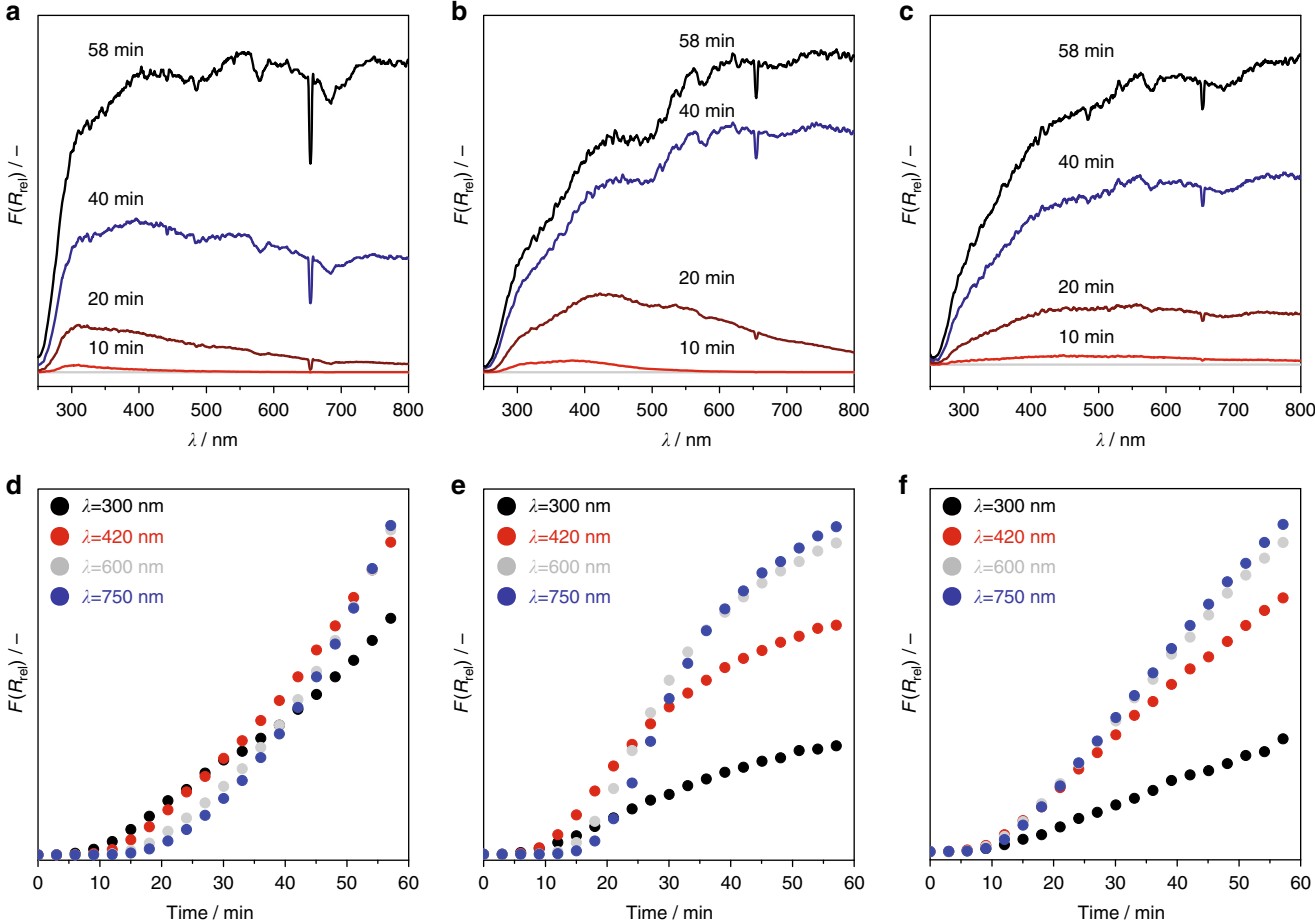

**Fig. 2** Coke formation in PDH. The UV-vis spectra of $ZrO_2$ with **a** 9.1, **b** 13.0 or **c** 43.4 nm crystallites after different times on PDH stream. **d–f** Temporal evolution of Kubelka–Munk function at 300 (black circle), 420 (red circle), 600 (light grey circle) and 750 nm (blue circle) under PDH conditions. PDH was performed at 550 °C using a feed containing 40 vol% $C_3H_8$ in $N_2$

interact with each other to initially form small aromatic structures followed by further oligomerization and condensation to large graphitic structures[15]. It is reasonable to suggest that propene molecules adsorbed horizontally to catalyst surface can recombine owing to their spatial location. Such adsorption should be more favourable for Zr sites located on flat $ZrO_2$ surfaces in comparison to those located on corners or edges. Upon increasing the size of crystallites, the fraction of the former species will decrease and thus will result in a higher rate of carbon deposition. Exactly this trend was observed experimentally. Further experimental and theoretical studies are, however, required to clarify the size effect on coke formation.

We turn our discussion back to catalyst activity to answer two important questions. Can $ZrO_2$ activate C–H bond in other alkanes? Is the size–activity relationship valid for PDH only? To this end, we performed catalytic tests with iso-butane, ethane and methane with the latter possessing the highest C–H bond strength among alkanes. Due to the thermodynamic constrains, methane activation was investigated at 800 °C, while the tests with other alkanes were carried out at 550 °C, where PDH was also performed. Ethylene was the only product observed in the tests with methane. It is, however, worth mentioning that due to the low $CH_4$ conversion of only 0.015%, it was difficult to precisely conclude if other products were also formed. Therefore, we do not discuss product selectivity in $CH_4$ conversion tests. On the basis of previous studies on the oxidative coupling of methane[26] and non-oxidative methane conversion to ethylene[27], $CH_3$ radical is suggested to be formed through breaking the C–H bond in

$CH_4$. Two such radicals recombine to $C_2H_6$ followed by its dehydrogenation to $C_2H_4$. $ZrO_2$ is actually able to catalyse the latter reaction (Fig. 3a).

As seen in Fig. 3a, the feed alkanes can be ordered in terms of the rate of olefin formation as follows $CH_4 < C_2H_6 < C_3H_8 <$ iso-$C_4H_{10}$. This activity order actually correlates with the strength of the C–H bond in these alkanes. The corresponding values of the weakest C–H bond in these alkanes are 439.3, 420.5, 410.5 and 400.4 kJ mol$^{-1}$. Regardless of the size of $ZrO_2$ crystallites, the activity order did not change, while the dehydrogenation rate of $C_2H_6$, $C_3H_8$ and iso-$C_4H_{10}$ increased with a decrease in the size (Fig. 3b). $CH_4$ conversion tests with $ZrO_2$ materials differing in the size of their crystallites were not performed because this alkane requires too high temperature, where different structural changes in $ZrO_2$ will occur.

**Benchmarking and demonstrating catalyst durability.** To demonstrate the potential of $ZrO_2$ for PDH from an applied viewpoint, we determined the space time yield of propene (STY ($C_3H_6$)) over the best performing catalyst from Fig. 1b at different temperatures and propane conversion above 25%. The selectivity to propene was above 85%. Carbon deposits were the main side product with the selectivity of about 10%. Although they are not desired, such deposits indirectly contribute to minimizing overall costs in the course of CATOFIN process through using the heat released upon their combustion during catalyst regeneration phase for the endothermic PDH reaction[14]. Figure 4a shows a

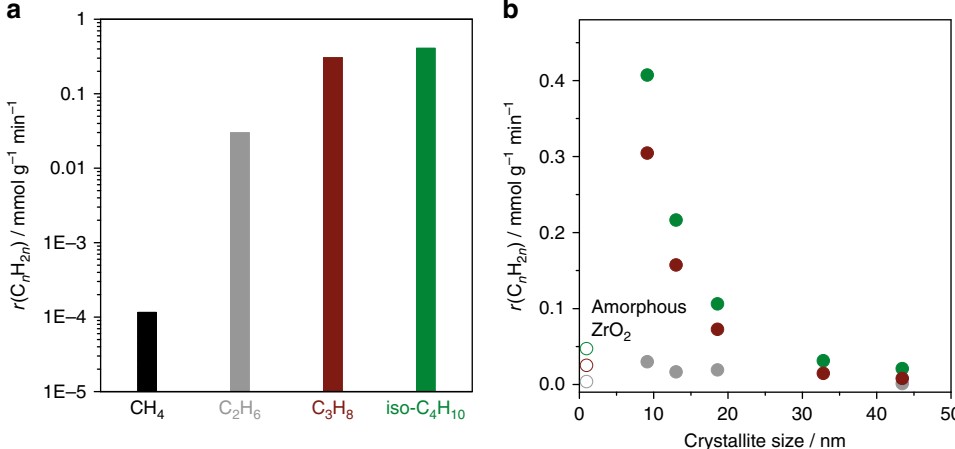

**Fig. 3** Catalytic activity for C–H bond breaking. The rate of olefin formation ($r(C_nH_{2n})$), i.e. $C_2H_4$, $C_2H_4$, $C_3H_6$ and iso-$C_4H_8$ from $CH_4$ (black), $C_2H_6$ (grey), $C_3H_8$ (dark red) and iso-$C_4H_{10}$ (green), respectively, **a** versus feed alkane over $ZrO_2$ with the highest $S(C_3H_6)$ in Fig. 1b or **b** versus the size of $ZrO_2$ crystallites. The rate values obtained over amorphous $ZrO_2$ in **b** are shown with open symbols. All these tests were performed using a feed containing 40 vol% alkane in $N_2$ at 550 °C for $C_2H_6$ (grey bar in **a** and grey circle in **b**), $C_3H_8$ (dark red bar in **a** and dark red circles in **b**), and iso-$C_4H_{10}$ (green bar in **a** and green circle in **b**) or at 800 °C for $CH_4$ (black bar in **a**)

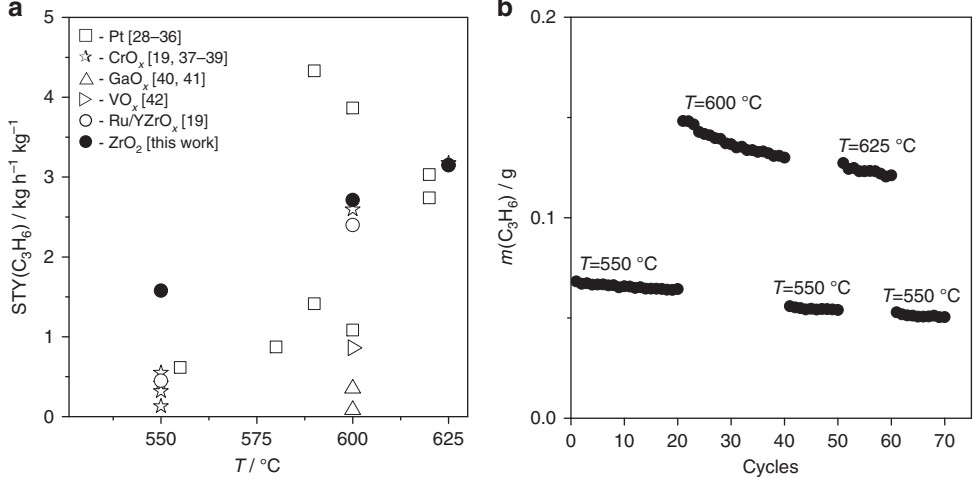

**Fig. 4** Comparison with literature data and durability. **a** Space time yield of propene ($STY(C_3H_6)$) obtained over $ZrO_2$ and the most active catalysts from previous studies[28-42] (Supplementary Table 3). The data at propane conversion above 25% were considered. **b** The amount of propene ($m(C_3H_6)$) formed within each 30 min PDH cycle in a series of 70 PDH/regeneration cycles at 550, 600 and 625 °C with WHSV of 1.57, 6.28 and 9.42 h$^{-1}$, respectively. $ZrO_2$ is the catalyst with the highest $S(C_3H_6)$ in Fig. 1b

comparison of the present $STY(C_3H_6)$ values with those obtained over state-of-the-art catalysts[28-42] (Supplementary Table 4). All traditional catalysts with supported species of different metal oxides including chromium are less active than $ZrO_2$. The latter also showed higher activity in comparison with doped $ZrO_2$-based catalysts possessing supported Ru. Even in comparison with highly active but expensive Pt-based catalysts, $ZrO_2$ proved its outstanding PDH activity.

Further practical relevance of $ZrO_2$ was demonstrated over about 13 days on stream in a series of 70 PDH/regeneration cycles at 550, 600 and 625 °C. No reductive treatment was performed before PDH cycles because the catalyst can generate $Zr_{cus}$ sites in situ through removal of lattice oxygen by propane/propene. The yield of propene within one PDH cycle initially increased and then decreased with rising time on stream at 550 °C but continuously decreased at higher temperatures (Supplementary Fig. 9). The increase is related to the formation of $Zr_{cus}$ sites, while the decrease is due to catalyst deactivation caused by

formation of carbon deposits, which could be removed upon oxidative catalyst regeneration as proven by stable catalyst performance within first 20 cycles at 550 °C (Fig. 4b).

When performing PDH at higher temperatures, the catalyst was less durable but showed high productivity. Such change of catalyst durability is due to increasing the size of crystallites caused by temperature-induced sintering process (Supplementary Fig. 10). Owing to small changes in the size, the catalyst was still active and durable at 550 °C (last 10 cycles in Fig. 4b) after the preceding PDH tests at 600 and 625 °C. Propene selectivity was about 90% (Supplementary Fig. 11).

**Role of $ZrO_2$ nanostructure for formation of oxygen defects.** High-angle annular dark-field scanning transmission electron microscopy (HAADF-STEM) was applied to characterize crystal morphology as a function of the size of crystallites. Figure 5a–c shows representative HAADF-STEM images of two crystalline

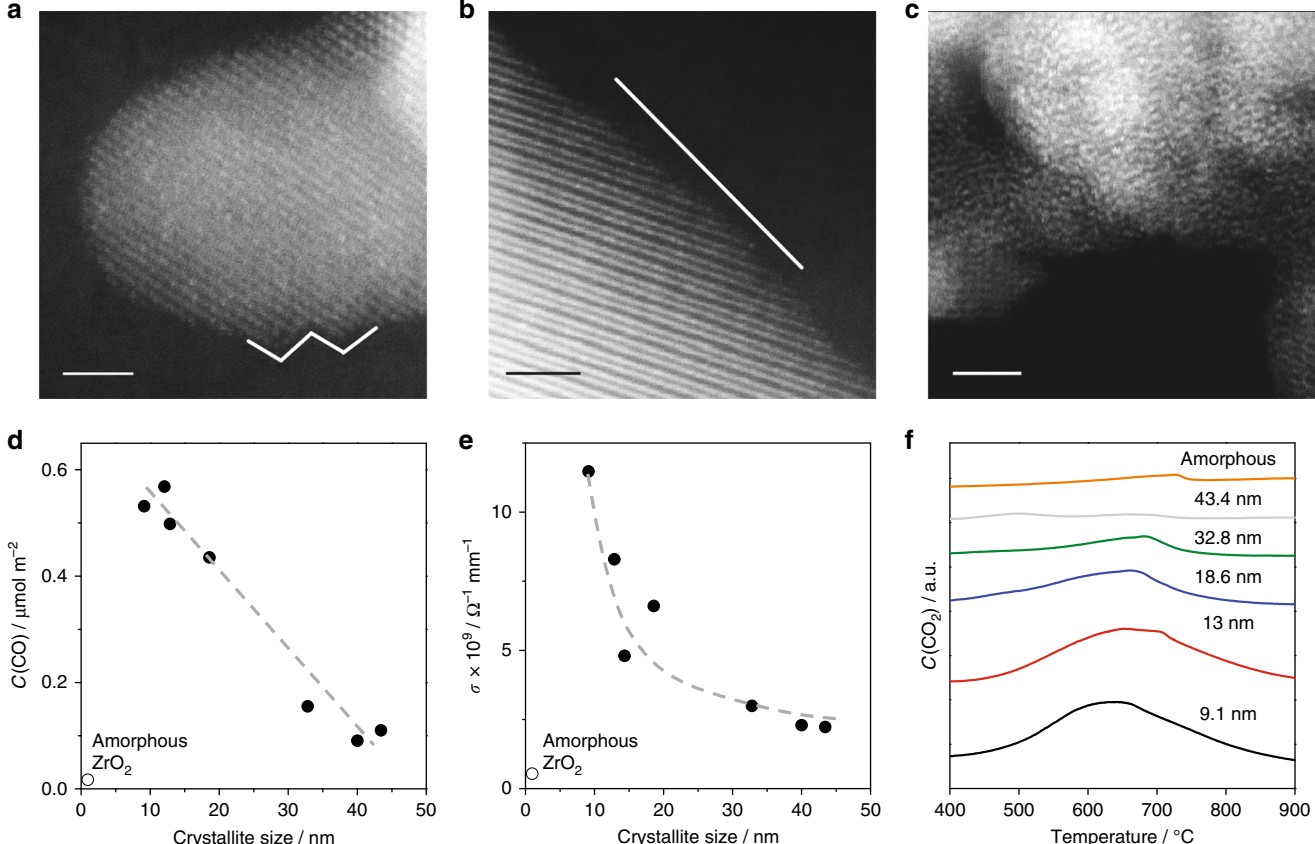

**Fig. 5** Morphology, acidic, conductivity and redox properties. HAADF-STEM images of $ZrO_2$ with crystallites of **a** 9.1 nm, **b** 43.4 nm or **c** amorphous $ZrO_2$. **d** Surface-normalized concentration of CO ($C(CO)$) determined from the amount of CO desorbed in CO-TPD. **e** Electrical conductivity at 550 °C in air. **f** Concentration of $CO_2$ ($C(CO_2)$) detected upon CO-TPR. Scale bars: 2 nm

samples with crystallites of 9.1 or 43.4 nm and one amorphous sample. Further representative images are given in Supplementary Fig. 12. There are significant differences between the samples in the structure of the surface depending on the crystallite size. Zigzag edges were seen in some images of the sample with 9.1 nm crystallites (Fig. 5a) and indicate the presence of a large number of corner atoms. Such surface defects should be coordinatively unsaturated zirconium and/or oxygen ions. They were not observed in the sample with 43.4 nm crystallites. Nearly perfect lattice planes with no corner atoms are typical for this sample (Fig. 5b), while no lattice planes are visible for the XRD-amorphous sample (Fig. 5c).

The HAADF-STEM technique can provide only limited quantitative information about the concentration and distribution of defect sites. In this regard, we applied surface-sensitive temperature-programmed desorption (TPD) of probe molecules differing in their Lewis basicity strength. For minimizing the effect of Brønsted sites on adsorption/desorption properties of $ZrO_2$[20], all TPD tests were carried out with the samples treated at 550 °C to remove acidic OH groups before adsorbing probe molecules.

CO naturally adsorbing on the $Zr_{cus}$ sites (Lewis acidic sites) desorbed between 250 and 450 °C (Supplementary Fig. 13). The surface-normalized concentration of sites for CO adsorption was established to increase with a decrease in the size of crystallites (Fig. 5d), while the temperature of maximal rate of CO desorption was not affected (Supplementary Fig. 13). The lowest amount was detected for amorphous $ZrO_2$. A similar effect of the crystallite size on the concentration of adsorption sites for $NH_3$

and $C_3H_6$ was also obtained when these compounds were used as probe molecules for titrating Lewis acidic sites (Supplementary Fig. 14). Regardless of the probe molecule the rate of alkene formation from $C_2H_6$, $C_3H_8$ and iso-$C_4H_{10}$ rose with an increase in the concentration of the accordingly determined Lewis acidic sites (Supplementary Fig. 15). Thus, the TPD results suggest that nanostructure is vital for the stabilization/formation of surface Lewis acidic defects.

We also applied electrical conductivity tests at 550 °C for semi-quantitative analysis of the number of bulk anion vacancies, which are responsible for oxygen-ionic conductivity in $ZrO_2$-based materials[43] and direct indicators for the presence of $Zr_{cus}$ sites. To check the contribution of electronic conduction, we analysed the effect of oxygen partial pressure on overall conductivity. Regardless of the size of crystallites, the conductivity decreased by only a factor of less than 2, which is lower than 21 as expected for a pure p-conductor upon reducing the pressure from 20 kPa to about $10^{-4}$ kPa (Supplementary Table 5). This result proves that all tested $ZrO_2$ samples mainly possess ionic conductivity. As seen in Fig. 5e the conductivity increased with a decrease in the size of crystallites. Such relationship is due to the higher concentration of anion vacancies or/and higher diffusivity of $O^{2-}$ in the samples with smaller crystallite size. It is worth mentioning that the conductivity of amorphous $ZrO_2$ was very low (Fig. 5e). Thus, an optimal size of $ZrO_2$ crystallites is required to achieve the highest conductivity.

The crystallite size is also decisive for the formation of anion vacancies as proven by TPR tests with CO (Fig. 5f). The smaller the size, the higher the amount of lattice oxygen was removed in

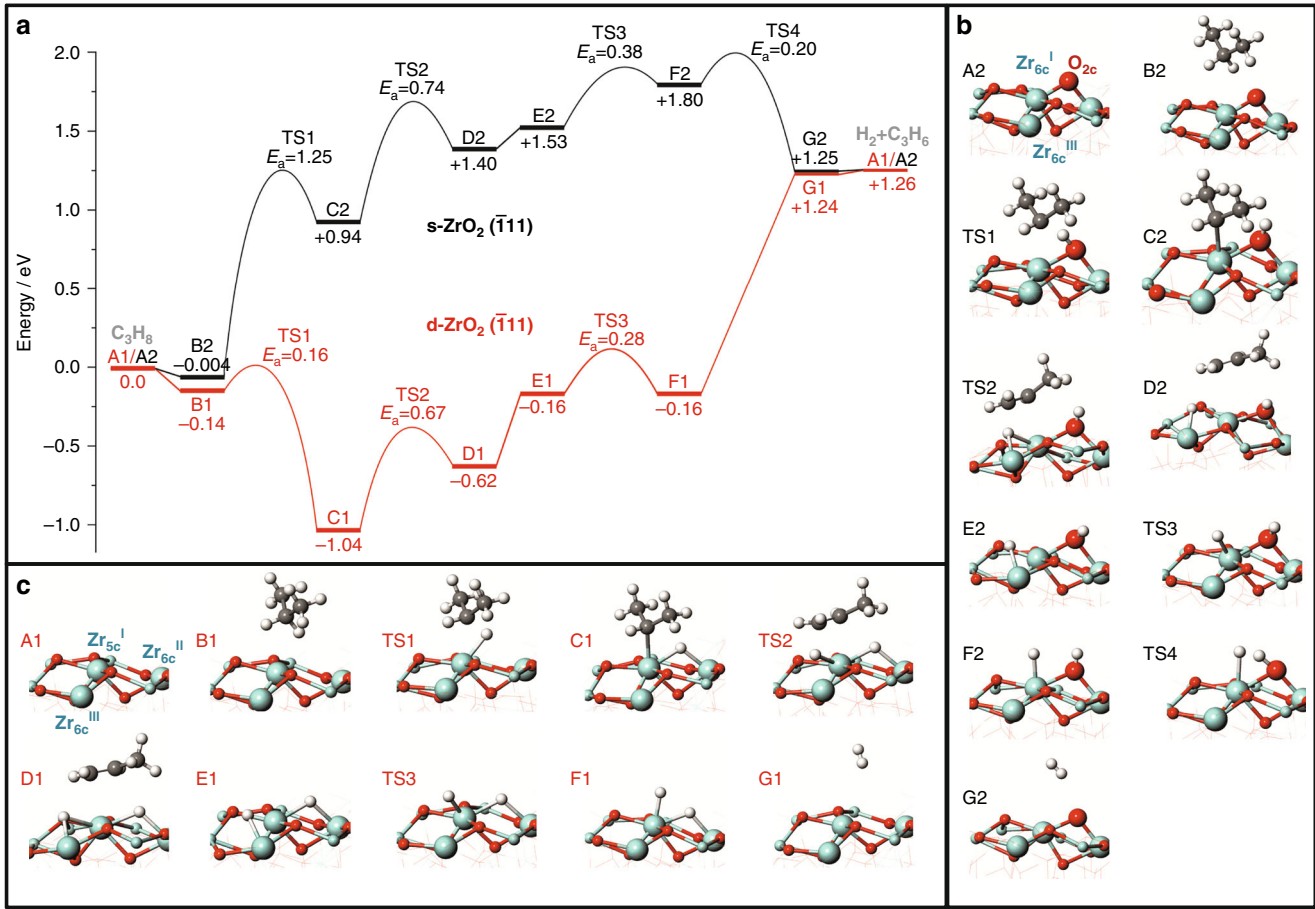

**Fig. 6** Mechanism of PDH. **a** The calculated energy profiles along the pathways of propane dehydrogenation to propene and the optimized structures of intermediates and transition states (TS) on **b** s-ZrO$_2$($\bar{1}$11) and **c** d-ZrO$_2$($\bar{1}$11) surfaces (Cyan, grey, red and white symbols stand for Zr, C, O and H, respectively)

form of CO$_2$. This result is in agreement with previous DFT calculations predicting that the energy for the formation of an oxygen vacancy in ZrO$_2$ lowers for nanoscale particles[44]. However, amorphous ZrO$_2$ produced very low amount of CO$_2$ from CO (Fig. 5f). Thus, there should be a minimal size of crystallites, below which the removal of lattice oxygen from ZrO$_2$ will be hindered.

**Molecular level details on C$_3$H$_8$ dehydrogenation over ZrO$_2$.** With the purpose to derive molecular insights into the kind of the catalytically active sites for propane dehydrogenation to propene over monoclinic ZrO$_2$, we performed DFT calculations. The focus was put on the role of Zr$_{cus}$ sites. As the ($\bar{1}$11) facet represents the most stable surface plane of monoclinic ZrO$_2$[45–47], it was used as the basis for computing stoichiometric (s-ZrO$_2$($\bar{1}$11)) and oxygen-defective (d-ZrO$_2$($\bar{1}$11)) surfaces. Computational details are given in Supplementary Fig. 16, Supplementary Table 3 and Supplementary Table 6. Although the energies for activating C$_3$H$_8$ at the methylene or methyl C–H bond to yield *iso*-C$_3$H$_7$ or *n*-C$_3$H$_7$ are close (Supplementary Table 7), subtraction of a second hydrogen to yield propene is kinetically more favourable for the *iso*-C$_3$H$_7$ fragment. This result is in agreement with previous DFT calculations of PDH over other metal oxides[21–23]. Therefore, the first step of propane activation in our calculations was cleavage of the methylene C–H bond. The full potential energy surfaces of the most preferred routes on both surfaces are shown in Fig. 6, while the Gibbs free energy profiles are given in

Supplementary Fig. 17. This figure shows that the apparent free energy barrier for propane dehydrogenation over the d-ZrO$_2$ ($\bar{1}$11) facet is 1.66 eV lower than that over the s-ZrO$_2$($\bar{1}$11) facet (2.52 vs. 4.18 eV) thus indicating that the dehydrogenation reaction is kinetically more favoured over the d-ZrO$_2$($\bar{1}$11) facet than over the s-ZrO$_2$($\bar{1}$11) facet.

Methylene C–H bond activation in propane: To identify the active sites and the modes of methylene C–H bond dissociation on the s-ZrO$_2$($\bar{1}$11) surface, we computed this route on a centre consisting either of one Zr site and one O site or two O sites. For the sake of completeness, six- and seven-fold coordinated Zr cations (Zr$_{6c}^{I}$, Zr$_{7c}^{II}$, Zr$_{6c}^{III}$ and Zr$_{6c}^{IV}$) as well as twofold and threefold coordinated oxygen anions (O$_{2c}$, O$_{3c}^{I}$, O$_{3c}^{II}$, O$_{3c}^{III}$ and O$_{3c}^{IV}$) were considered (Supplementary Fig. 16). A Zr–O pair was found to be the most favourable centre to break the methylene C–H bond through a heterolytic route (Supplementary Fig. 18 and Supplementary Table 7). The Zr$_{6c}^{I}$–O$_{2c}$ site should possess the highest reactivity because of the formation of the most stable Zr$_{6c}^{I}$–*iso*-C$_3$H$_7$ and O$_{2c}$–H intermediates owing to the stronger basicity of O$_{2c}$ in comparison with O$_{3c}$. This step is endothermic by 0.94 eV and has an activation barrier of 1.25 eV (Fig. 6).

In contrast to s-ZrO$_2$($\bar{1}$11), methylene C–H bond cleavage over d-ZrO$_2$($\bar{1}$11) is a homolytic process without involving lattice oxygen. This reaction should occur on two adjacent Zr cations, *i.e.* Zr$_{5c}^{I}$ and Zr$_{6c}^{II}$ ([Zr$_{5c}^{I}$, Zr$_{6c}^{II}$]–O$_{v}$) or Zr$_{5c}^{I}$ and Zr$_{6c}^{III}$ ([Zr$_{5c}^{I}$, Zr$_{6c}^{III}$]–O$_{v}$), and is exothermic by 1.04 or 0.55 eV (Fig. 6 and Supplementary Table 8). The Zr$_{5c}^{I}$ and Zr$_{6c}^{II}$ cations are located at an anion vacancy (O$_{v}$), while Zr$_{6c}^{III}$ is a regular surface site.

The formed $iso$-$C_3H_7$ fragment is bound to the $Zr_{5c}^I$ site, while the H atom is located in a bridged position between $Zr_{6c}^{II}$ or between $Zr_{5c}^I$ and $Zr_{6c}^{III}$. On the basis of the activation barrier (0.16 vs. 0.73 eV) for breaking the methylene C–H bond (Supplementary Fig. 19), the $[Zr_{5c}^I, Zr_{6c}^{II}]$–$O_v$ site should be more reactive than the $[Zr_{5c}^I, Zr_{6c}^{III}]$–$O_v$ site. With respect to the $Zr_{6c}^I$–$O_{2c}$ site on the s-$ZrO_2(\bar{1}11)$ surface, the activation barrier for this reaction pathway decreases by 1.09 eV in the presence of oxygen vacancy (Fig. 6) thus strongly facilitating propane activation.

To check the importance of oxygen defects for activation of methane, which is the most inert alkane, we computed breaking C–H bond over the s-$ZrO_2(\bar{1}11)$ and d-$ZrO_2(\bar{1}11)$ surfaces. Similarly to propane activation, $Zr_{6c}^I$–$O_{2c}$ and $[Zr_{5c}^I, Zr_{6c}^{II}]$-$O_v$ should be the corresponding active sites. The presence of oxygen defect was established to be essential for breaking the C–H bond in $CH_4$, too. The activation barrier on the $Zr_{6c}^I$–$O_{2c}$ site is 1.14 eV but is only 0.10 eV on the $[Zr_{5c}^I, Zr_{6c}^{II}]$-$O_v$ site. Moreover, $CH_4$ activation on the latter centre is exothermic by 1.06 eV.

$C_3H_6$ and $H_2$ formation: After having identified the first step of propane activation over s-$ZrO_2(\bar{1}11)$, we examined transformation routes of the $Zr_{6c}^I$–$iso$-$C_3H_7$ and H–$O_{2c}$ intermediates. The most thermodynamically and kinetically preferred way for $Zr_{6c}^I$–$iso$-$C_3H_7$ conversion is the cleavage of the methyl C–H bond at the $Zr_{6c}^{III}$ site (Supplementary Fig. 20). It is an endothermic process (0.46 eV) and has an activation barrier of 0.74 eV (C2 to D2 in Fig. 6), which is 0.51 eV lower than that for the activation of the first methylene C–H bond in propane. The so-formed propene adsorbs weakly (0.13 eV) at the $Zr_{6c}^I$ site, while the co-generated H fragment is bound to the neighbouring $Zr_{6c}^{III}$ site (Fig. 6).

When one H atom is subtracted from the methyl group of $Zr_{5c}^I$–$iso$-$C_3H_7$ stabilized on the d-$ZrO_2(\bar{1}11)$ surface (Fig. 6 and Supplementary Fig. 21), weakly adsorbed propene (0.46 eV) is formed (TS2, Fig. 6). The remaining H atom is bound to the $Zr_{5c}^I$ and $Zr_{6c}^{III}$ cations. This process is endothermic by 0.42 eV and has an activation barrier of 0.67 eV, which is slightly lower than that (0.74 eV) required for the activation of $Zr_{6c}^I$–$iso$-$C_3H_7$ over the stoichiometric surface. In addition to the mechanism and the kinetics of propene formation, the presence of anion vacancies also affects $H_2$ formation (Fig. 6). This process on the s-$ZrO_2(\bar{1}11)$ surface has an effective activation barrier of 0.47 eV and is exothermic by 0.28 eV, while the formation of $H_2$ on the d-$ZrO_2(\bar{1}11)$ surface is endothermic by 1.40 eV ($H_2$ dissociation is barrierless).

## Discussion

The work presented here explains at molecular level the importance of oxygen defects on the surface of $ZrO_2$ for efficient C–H bond activation in $C_1$–$C_4$ alkanes. Owing to the presence of such defects, this typically non-reducible metal oxide showed unexpectedly high activity in the non-oxidative propane dehydrogenation to propene and outperformed the state-of-the-art catalysts possessing supported metal oxides or platinum. Its unique performance is related to the presence of surface coordinatively unsaturated Zr cations ($Zr_{cus}$) located at anion vacancies in the lattice of $ZrO_2$. Our DFT calculations predict that $Zr_{cus}$ sites open an alternative pathway of propene and hydrogen formation with a lower activation barrier in comparison with non-defect $ZrO_2$. Two $Zr_{cus}$ sites directly participate in homolitic C–H bond activation, while lattice oxygen and zirconium cation are required for this process on the stoichiometric surface of $ZrO_2$. Although the concentration of $Zr_{cus}$ can be adjusted in a controlled manner through the size of $ZrO_2$ crystallites, further

improvements are expected when the coordination number of zirconium cations, their location on certain facets and/or the shape of crystallites can also be tuned. Our approach presented herein can be extended to other bulk oxides of non-reducible metals and offers the opportunity for tailoring their acid–base and redox properties as well as electrical conductivity for specific applications.

## Methods

**Catalyst preparation**. $ZrO_2$ samples were prepared by hydrothermal, precipitation, sol-gel, hard-template methods or simple decomposition of different zirconium salts. The used chemicals and selected physico-chemical properties of the synthesized materials are given in Supplementary Table 1. Further preparation details are provided in Supplementary Note 1.

**Catalyst characterization**. XRD powder patterns were recorded on a Panalytical X'Pert diffractometer equipped with a Xcelerator detector, automatic divergence slits and Cu tube (kα1/α2 radiation, 40 kV, 40 mA, $\lambda$ = 0.015406 nm, 0.0154443 nm). Cu beta-radiation was excluded by using nickel filter foil. The measurements were performed in 0.0167° steps and 25 s of data collecting time per step. Peak positions and profile were fitted with Pseudo-Voigt function using the HighScore Plus software package (Panalytical). The PDF-2 database of the International Centre of Diffraction Data (ICDD) was used for phase identification. The integral breadth method using the Scherrer equation under the assumption of spherically shaped crystallites was applied for calculating crystallite size from the ($\bar{1}11$) and (111) reflection peak and the average values are reported. The K parameter was set to 1.0747. The adjustment of the used diffractometer is weekly checked according to the Si (111) reflection peak at 28.433°. The deviation of the peak position is within the standard deviation (0.004°). We have also measured XRD of five randomly taken probes of $ZrO_2$ with an estimated crystallite size of 9.1 nm, which is one of the lowest values. The size of crystallites was determined with an error of 0.02 nm. Thus, the error in determining the size of crystallites is below 1%.

Transmission electron microscopy measurements were carried out at 200 kV with an aberration-corrected JEM-ARM200F (JEOL, Corrector: CEOS). The aberration-corrected STEM imaging (High-Angle Annular Dark Field) was performed with a spot size of approximately 0.1 nm, a convergence angle of 30–36° and collection semi-angle of 90–170 mrad.

Acid–base, redox and electrical properties of selected samples were determined from temperature-programmed desorption tests using CO, $NH_3$ or $C_3H_6$ as probe molecules, temperature-programmed reduction tests with $H_2$ or CO and electrical conductivity measurements respectively. Technical details of these tests are given in Supplementary Note 2.

In situ UV-vis tests were performed at 550 °C in an in-house developed set-up equipped with an AVASPEC fibre optical spectrometer (Avantes), a DH-2000 deuterium-halogen light source and a CCD array detector. A high-temperature reflection UV-vis probe consisting of six radiating optical fibres and one reading fibre was threaded through the furnace to face quartz reactor walls at the position where catalysts were held[48]. A feed containing 40 vol% $C_3H_8$ in $N_2$ was used in these tests. Initial propane conversion was about 10%.

For analysing carbon deposition during PDH, we defined relative reflectance ($R_{rel}$) as given in Eq. (1) as the ratio of the reflectance of the catalysts with coke deposits ($R_{DH}$) to the fully oxidized ones ($R_{O_2}$). From this reflectance, we calculated the relative Kubelka–Munk function $F(R_{rel})$ according to Eq. (2).

$$R_{rel} = \frac{R_{DH}}{R_{O_2}} \tag{1}$$

$$F(R_{rel}) = \frac{(1 - R_{rel})^2}{2 \times R_{rel}} \tag{2}$$

**Catalytic tests**. Catalytic tests with $C_2H_6$, $C_3H_8$ and $iso$-$C_4H_{10}$ were performed at 1 bar between 550 and 625 °C using an in-house developed set-up consisting of 15 continuous-flow fixed-bed quartz tubular (length and inner diameter are 465 and 4 mm, respectively) reactors operating in parallel. Tests with $CH_4$ were carried out at 800 °C in an in-house developed set-up consisting of six continuous-flow fixed-bed quartz tubular (length and inner diameter are 330 and 4 mm respectively) reactors operating in parallel. Depending on specific purposes, catalytic tests were performed with oxidized (treated in air at the reaction temperature) or reduced (treated in either $CO/N_2$ = 57/43 or $H_2/N_2$ = 57/43 at the reaction temperature) catalysts. Feeds containing 40 vol% $C_2H_6$, 40 vol% $C_3H_8$, 40 vol% $iso$-$C_4H_{10}$ or 40 vol% $CH_4$ in $N_2$ were used. The feed components and the reaction products were analysed by an on-line gas chromatograph (Agilent 6890) equipped with PLOT/Q (for $CO_2$), AL/S (for hydrocarbons) and Molsieve 5 (for $H_2$, $O_2$, $N_2$ and CO) columns as well as flame ionization and thermal conductivity detectors.

Further details about catalytic tests as well as formulae for calculating various catalyst characteristics are provided in Supplementary Note 3.

**DFT calculations**. Spin-polarized and periodic density functional theory (DFT) calculations were carried out by using the Vienna ab initio simulation package (VASP)[49,50]. Exchange and correlation were treated within the Perdew–Burke–Ernzerhof generalized gradient approximation (GGA-PBE)[51]. To obtain accurate energies with errors of less than 1 meV per atom, a cut-off energy of 400 eV was used. Geometry optimization was converged until forces acting on atoms were lower than 0.02 eV/Å, whereas the energy threshold defining self-consistency of the electron density was set to $10^{-4}$ eV. The climbing image nudged elastic band (CI-NEB) method with eight images was applied for finding transition states and minimum energy paths of all reactions[52]. The final transition state structures were refined by using the quasi-Newton algorithm until the Hellman–Feynman forces on each ion were lower than 0.02 eV/Å. The normal mode frequency analysis was performed to validate the optimized transition states and each authentic transition state has only one imaginary frequency along the reaction coordinates. The resulted zero-point vibrational energies (ZPE) from the frequency analysis are included in our energetic comparison and discussion. For the optimization of the bulk structure, the lattice parameters of the monoclinic $ZrO_2$ ($m$-$ZrO_2$) phase were determined by minimizing the total energy of the unit cell by using a conjugated gradient algorithm to relax the ions. A 7×7×7 Monkhorst−Pack k point grid was used for sampling the Brillouin zone[53]. Additional details are provided in Supplementary Methods. In addition, we also tested the corrections of Hubbard term (PBE + U) and dispersion (DFT + D3). These new data are now presented in Supplementary Fig. 22 and Supplementary Fig. 23.

## Data availability

The authors declare that the data supporting the findings of this study are available within the paper and its supplementary information. Further information is also available from the corresponding authors upon reasonable request.

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

## Acknowledgements

Financial support by Deutsche Forschungsgemeinschaft (KO 2261/8-1, JI 210/1-1), the National Natural Science Foundation of China (Grants 91645108, U1162117), Science Foundation of China University of Petroleum, Beijing (C201604) and the State of Mecklenburg-Vorpommern are gratefully acknowledged. Ya.Z. acknowledges support from the China Scholarship Council.

## Author contributions

E.V.K. conceived and coordinated all stages of this research. All authors participated in planning, discussing and interpreting the experimental and theoretical data as well as in writing and revising the manuscript. Various catalytic and characterization experiments as well as DFT calculations were performed by Ya.Z., Yu.Z., T.O., H.L., M.-M.P. and H.J., who also evaluated the data obtained.

## Additional information

**Competing interests:** The authors declare no competing interests.

