## [Peer Review File · Nature Communications]

Reviewers' comments:

Reviewer #1 (Remarks to the Author):

The work describes the dehydrogenation of propane on ZrO₂ and the implication of coordinatively unsaturated sites as the active sites for propane dehydrogenation

While interesting this study suffers from many shortcomings:

- 1) Low activity compared to Cr or Pt based systems
- 2) Low stability (which is the major issue in all PDH catalyst)
- 3) Lack of discussion of literature precedent that describes very similar findings on the same or related materials
- 4) Activity should be presented not as a function of particle size but per surface area (and not activity per surface area as a function of particle size)
- 5) The potential energy surface displays very high barrier of energy, which indicates that the reaction should not take place at 600 °C. In addition, the reaction energy path should be given in Free Energy (which will make things even worse)

Overall the results are too preliminary

Reviewer #2 (Remarks to the Author):

This work titled “Control of coordinatively unsaturated Zr sites in ZrO₂ for efficient C-H bond activation” presents some interesting results as to how the crystallite size can help us to modify the concentration of coordinatively unsaturated Zr sites which are active sites for propane dehydrogenation. The bare ZrO₂ having higher oxygen vacancies showed better activity for PDH reaction compared to some state-of-the-art catalysts. The findings are novel and original and may be of interest to other community. However, it lacks some experimental data to validate their claims and could be considered for publication after major revision. The following points must be addressed by authors before publication:

1. The title of the manuscript indicates that the method presented is a generic one for C-H bond activation. However, C-H bond activation is only shown for propane in the PDH reaction and a brief theoretical discussion for methane. Either, authors should conduct more experiments/theory

with different alkanes to show C-H bond activation for a range of reactions or the title should be specific for PDH reaction. In that case the discussion on methane could be more suited in supporting information.

2. The mechanism (the coordinatively unsaturated Zr sites) needs to be supported by the EXAFS analysis. How does the coordination of zirconium is influenced by the crystallite size needs to be discussed with EXAFS results which will give information about local structure, coordination geometry of Zr. It is not clearly explained how the coordinatively unsaturated Zrcus present in the smaller crystallites (but not in the amorphous oxide which may have more coordinatively unsaturated Zr) are responsible for the higher activity of PDH reaction.

3. Quantification of Zr(cus)/oxygen vacancies by oxygen pulse analysis is not done here though the authors used it in their previous works. Since the authors mentioned that the PDH activity is governed by number of active sites(in the crystallites) it is important to quantify it for comparison.. What is the critical crystallite size beyond which the activity decreases (when the size reduces further, going towards amorphous).

4. The zig-zag edges for the smaller crystallites is not everywhere(Fig 3a) and it will be a qualitative interpretation to link this observation to a large number of corner atoms in smaller crystallites.

5. The CO – TPD/ TPR and the conductivity tests for amorphous zirconia needs to be compared with the crystallite zirconia in the characterisation.

6. Interestingly, propene selectivity also increased with decreasing crystallite size (Figure 1b). What could be the reason for this? Is it possible to have experimental or theoretical insight on this?

7. How does crystallite size affects the activation energy of the reaction? How does the Arrhenius factor contributes to the rate of the reaction at smaller crystallite size ?

8. Will it work with other oxides like Titania? or it is specific to zirconia alone?

Reviewer #3 (Remarks to the Author):

The manuscript entitled with " Control of coordinatively unsaturated Zr sites in ZrO₂ for efficient C-H bond activation" presents original and interesting data. The importance of propane dehydrogenation reaction and reducible oxides for chemical industry and heterogeneous catalysis

warrants for wide interest across a variety of scientific communities. The experiments presented in this paper have been very carefully executed and analyzed. However, given their impact in the field, I believe further refinement in their interpretation and discussion is needed before publication in Nature Communication. Here are specific questions:

Detailed points:

- (1) Previous works have investigated the effect of crystalline sizes on Metal-Oxygen (M-O) coordination and the C-H activation. The results showed that metal oxides with larger particle sizes have longer M-O bonds, leading to the favorable formation of uncoordinated M sites (J. Catal. 293, 175–185, ACS Catal. 7, 4, 2419-2424), which is contrast to this work.
- (2) The bulk ZrO₂ showed decreasing C₃H₆ formation and increasing carbon deposit with time. Due to C₃H₈, C₃H₆ and H₂ can serve as reductive gases, how about the structural evolutions of such ZrO₂ under operating conditions.
- (3) The authors present the mean crystalline sizes of ZrO₂ by XRD, however the morphologies and sizes are not uniform. Therefore, correlating C₃H₆ formation with crystalline sizes directly may be not correct. Moreover, the authors should present the crystalline sizes with error bars. In addition, what about the XRD and Raman peak shifts when oxygen vacancies were generated?
- (4) Previous work (Catal. Sci. Technol., 2017, 7, 4499-4510) has shown that the ZrO₂ with small crystalline sizes present more acidic sites, which bind C₃H₆ strongly and thus lead to further dehydrogenation to form coke. However, why did the smaller ZrO₂ with more acidic sites show lower coke selectivity in this work? Also, smaller ZrO₂ should show less sintering-resistant ability during dehydrogenation and redox cycles, which differs from this work.
- (5) The authors quantified the surface oxygen vacancies using TPD measurement, how about the XPS results?
- (6) The authors prepared different sizes of ZrO₂ through different methods and at different calcination temperatures, even much lower than the reaction temperatures. Is there any structural changes of the low-temperature-prepared catalysts during reactions?

Reply to the comments to NCOMMS-18-08612-T

The scientific comments from the reviewers are highly appreciated by the authors. The manuscript has been revised accordingly. Our detailed replies to the specific comments are given below.

Reply to Reviewer 1

Comment 1

Low activity compared to Cr or Pt based systems

Reply to comment 1

Figure 2 in the original manuscript (Figure 4 in the emended manuscript) shows the space time yield (STY) of propene formation over our ZrO₂ and the most active, state of the art catalysts including Cr or Pt containing materials. One can clearly see that only two Pt-based catalysts among 17 previously tested catalysts showed higher STY than our catalyst. The latter material is pure ZrO₂ without any metal or metal oxide possessing dehydrogenation activity. STY values between 1.6 and 3.1 kg(C₃H₆)/h/kg(catalyst) achieved in the present study are attractive from an application point of view.

In addition, we revealed the molecular structure of catalytically active sites and provided fundamental principles controlling their concentration. This knowledge opens a possibility to tune activity of ZrO₂-based catalysts and can be used for preparation of other materials with high activity in alkane dehydrogenation.

Comment 2

Low stability (which is the major issue in all PDH catalyst)

Reply to comment 2

We agree with the reviewer that our catalysts deactivate with time on propane stream but such behaviour is also valid for commercial catalysts, as actually the reviewer wrote "... which is the major issue in all PDH catalyst". A commercial Cr-based catalyst operates for about 10 minutes followed by oxidative regeneration. Depending on reaction temperature, ZrO₂ performs more or less stable for up to 20 min. Its activity can be restored after oxidative catalyst treatment at the dehydrogenation temperature as proven in a series of 70 dehydrogenation/regeneration cycles lasted over 13 days on stream. In comparison to commercially applied Pt-based catalysts, ecologically harmful Cl₂ or Cl-containing compounds are not required to regenerate our catalyst.

Comment 3

Lack of discussion of literature precedent that describes very similar finding on the same or related materials

Reply to comment 3

We are quite unsure what the reviewer means. To the best of our knowledge there are only a few papers dealing with alkane dehydrogenation over bulk ZrO₂-based materials without supported Pt or MO_x (M=V, Cr, Ga, Mo) species, which are actively participating in the reaction. Our group was actually first who reported about such novel catalysts. We have written in the introduction of the original manuscript "Recently, we developed eco-friendly and cost-efficient catalysts on the basis of ZrO₂, which had to be promoted with metal oxides and contain tiny

amounts of supported Ru or Rh NP to show high activity^{18,19}. Coordinatively unsaturated Zr (Zr_{cus}) and neighbouring lattice oxygen were suggested to participate in PDH²⁰. Yet, the promoter used to purposefully create Zr_{cus} sites either increased or decreased the activity hence proving the limitations of this classical approach for tailored catalyst design. Here, we describe how nanostructuring of ZrO_2 crystallites enables the identification of the nature of active sites and the control of their concentration without the usage of any dopant or supported species.”

The present study deals with non-doped ZrO_2 materials, which have been never published before. In comparison to our previous studies, we provide new fundamental insights into the role of Zr_{cus} sites. “A structural model of the active site was established owing to our multidisciplinary approach combining density functional theory (DFT) calculations with a number of complementary experimental methods including catalytic tests. The active site consists of two Zr cations located at an oxygen vacancy, which homolytically break the C-H bond in alkanes. The kind of this site differs from that previously suggested by some of us for doped ZrO_2 -based catalysts^{18, 19} and by other researchers for different metal oxides used for PDH²¹⁻²³, where metal cation and neighbouring lattice oxygen were suggested to form the active site. Bare ZrO_2 designed especially to maximize the concentration of the active sites showed industrially relevant performance in comparison with commercial-like catalysts containing CrO_x or Pt species and to other alternative state-of-the-art catalysts.”

Comment 4

Activity should be presented not as a function of particle size but per surface area (and not activity per surface area as a function of particle size).

Reply to comment 4

We cannot fully follow the reviewer’s criticism. According to her/his suggestion, the rate (probably related to catalyst amount) will simply increase with the surface area. Which important scientific information can be derived from such presentation? How will such dependence help to understand how and if the surface area affects catalyst activity?

Our experimental data were actually reported in form of propene formation rate related to either catalyst amount (Fig. 1a) or specific surface area (Supplementary Fig. 2) The latter form is typically used to take into account the effect of specific surface area of different heterogeneous catalysts on their activity. Regardless of the rate expression form, catalyst activity increases with a decrease in the size of crystallites thus proving that the size but not only the surface area determines the activity.

Nevertheless, as the reviewer suggested we have prepared two additional figures showing the rate of propene formation related to catalyst amount or specific surface area versus the surface area. The latter dependence clearly demonstrates that the surface area is not the only factor affecting catalyst activity. To further support our statement, we have provided an additional figure to the Supplementary information (Supplementary Fig. 2). This figure shows apparent activation energy of propene formation over ZrO_2 differing in the size of crystallites. The energy increases from about 140 to 300 $kJ \cdot mol^{-1}$ with an increase in the size from 9 to 43 nm.

The manuscript has been emended on page 5 as follows.

“The obtained relationship between the rate and the size of crystallites is also valid when the rate is calculated with respect to catalyst surface area (Supplementary Fig. 2a). Thus, the PDH activity of ZrO_2 should be governed by the number of the active sites. A simple effect of the crystallite size on the activity can be excluded, as the rate related to catalyst surface area increases with the area (Supplementary Fig. 2b-c). No effect of the area would be visible if this

catalyst property determined the activity exclusively. A further experimental support for the importance of the size of crystallites for PDH is the fact that apparent activation energy of propene formation depends on the size. The energy increases from about 140 to 300 kJ·mol⁻¹ with an increase in the size from 9 to 43 nm (Supplementary Fig. 3). This effect is related to the kind of catalytically active sites as will be demonstrated below when taking further catalytic and characterisation studies as well as DFT calculations into account.”

Comment 5

The potential energy surface display very high barrier of energy, which indicate that the reaction should not take place 600°C. In addition, the reaction energy path should be given in Free Energy (which will make things even worth)

Reply to comment 5

As the reviewer suggested, we have calculated the Gibbs free energies and total energies of all relevant states (including transition states) along the reaction coordinate. The corresponding diagram has been added to the Supplementary information (Supplementary Fig. 17). However, it should be noted that both propane and propene do not follow the ideal gas rule and this makes such a pressure and temperature dependent reaction more complicated than expected on the basis of the ideal gas models. Therefore, the results in Supplementary Fig. 17 are to be used only for qualitative comparison. Nevertheless, the calculations predict that the defect surface is more favourable for PDH than its ideal counterpart.

In addition, we also tested the corrections of dispersion (DFT+D3) and Hubbard term (PBE+ U_{eff}). These new data are now presented in the Supplementary Materials (Supplementary Fig. 22 and Supplementary Fig. 23).

It is not obvious that high energy barrier means that the reaction will not take place.

General statement

Overall the results are too preliminary

Reply to the general statement

This statement is surprising for us because the reviewer did not explain why the results are preliminary when particularly taking into account our multidisciplinary approach combining various complementary experimental techniques with density functional theory calculations. This combination enabled us to conclude, for the first time, what the active sites for alkane dehydrogenation over ZrO₂ are and how their concentration can be tuned. The kind of this site differs from that previously suggested by some of us for doped ZrO₂-based catalysts and by other researchers for different metal oxides used for propane dehydrogenation, where metal cation and neighbouring lattice oxygen form the active site.

Reply to Reviewer 2

Comment 1

The title of the manuscript indicates that the method presented is a generic one for C-H bond activation. However, C-H bond activation is only shown for propane in the PDH reaction and a brief theoretical discussion for methane. Either, authors should conduct more experiments/theory with different alkanes to show C-H bond activation for a range of reactions or the title should be specific for PDH reaction. In that case the discussion on methane could be more suited in supporting information.

Reply to comment 1

To strengthen our conclusion about the importance of Zr_{cus} sites for C-H bond activation, we have performed additional catalytic tests with C_2H_6 and iso- C_4H_{10} at $550^\circ C$ and with CH_4 at $800^\circ C$. The obtained results have been added to the emended manuscript (Fig. 3 and Supplementary Fig. 15). A clear dependence between the size of crystallites and the rate of dehydrogenation of C_2H_6 , C_3H_8 and iso- C_4H_{10} was obtained. As ZrO_2 will be restructured at $800^\circ C$, the effect of the size on the rate of CH_4 conversion was not investigated.

The manuscript has been emended on pages 9-10 as follows.

“We turn our discussion back to catalyst activity to answer two important questions. Can ZrO_2 activate C-H bond in other alkanes? Is the size-activity relationship valid for PDH only? To this end, we performed catalytic tests with iso-butane, ethane and methane with the latter possessing the highest C-H bond strength among alkanes. Due to the thermodynamic constrains, methane activation was investigated at $800^\circ C$, while the tests with other alkanes were carried out at $550^\circ C$, where PDH was also performed. Ethylene was the only product formed from CH_4 at a conversion degree of only 0.015%. As seen in Fig. 3a, the feed alkanes can be ordered in terms of the rate of olefin formation as follows $CH_4 < C_2H_6 < C_3H_8 < iso-C_4H_{10}$. This activity order actually correlates with the strength of the C-H bond in these alkanes. The corresponding values of the weakest C-H bond in these alkanes are 439.3, 420.5, 410.5 and $400.4 \text{ kJ}\cdot\text{mol}^{-1}$. Regardless of the size of ZrO_2 crystallites, the activity order did not change, while the dehydrogenation rate of C_2H_6 , C_3H_8 and iso- C_4H_{10} increased with a decrease in the size (Fig. 3b). CH_4 conversion tests with ZrO_2 materials differing in the size of their crystallites were not performed because this alkane requires too high temperature, where different structural changes in ZrO_2 will occur.”

The manuscript has been emended on pages 13-14 as follows.

Regardless of the probe molecule the rate of alkene formation from C_2H_6 , C_3H_8 and iso- C_4H_{10} rose with an increase in the concentration of the accordingly determined Lewis acidic sites (Supplementary Fig. 15).

Comment 2

The mechanism (the coordinatively unsaturated Zr sites) needs to be supported by the EXAFS analysis. How does the coordination of zirconium is influenced by the crystallite size needs to be discussed with EXAFS results which will give information about local structure, coordination geometry of Zr. It is not clearly explained how the coordinatively unsaturated Zr_{cus} present in the smaller crystallites (but not in the amorphous oxide which may have more coordinatively unsaturated Zr) are responsible for the higher activity of PDH reaction.

Reply to comment 2

We agree with the reviewer that EXAFS is a powerful technique for determining the chemical state of elements and to study the atomic and electronic structure of various materials.

Unfortunately, we do not have a direct access to in situ EXAFS measurements. We have also checked literature on this subject. We are not sure if it will be possible to analyse how the coordination of zirconium depends on the size of crystallites. For example, Rush et al. (J. Phys. Chem. B 104 (2000) 9597) wrote “Zr and Y K edge EXAFS spectra for the YSZ films with grain sizes of 6, 15, and 240 nm showed no major differences with the corresponding spectra of the bulk counterpart. This is clear proof that these nanocrystalline films exhibit similar levels of disorder to that of large crystals.” We are also quite unsure if EXAFS measurements will be surface sensitive when taking into account that the number of oxygen vacancies formed upon catalyst treatment in H₂ at 550 °C is very low compared to the number of lattice oxygen (1·10¹⁶ vs. 1·10²² per gram). To draw a definitive conclusion on this topic, further deeper EXAFS studies are required but are certainly behind the scope of our present study.

Our temperature-programmed tests with CO, NH₃, and C₃H₆ proved the presence of Zr_{cus} sites on the surface of ZrO₂ samples. All these molecules adsorb on Lewis acidic sites, i.e. Zr_{cus}. The importance of such sites for breaking the C-H bond is supported by the results shown in Fig. 1c, Supplementary Fig. 15, and most importantly by DFT calculations in Fig. 6 (Fig. 4 in the original manuscript).

Comment 3

Quantification of Zr_(cus)/oxygen vacancies by oxygen pulse analysis is not done here though the authors used it in their previous works. Since the authors mentioned that the PDH activity is governed by number of active sites(in the crystallites) it is important to quantify it for comparison. What is the critical crystallite size beyond which the activity decreases (when the size reduces further, going towards amorphous).

Reply to comment 3

The reviewer is right we have used O₂-titration tests for quantifying the number of anion vacancies in our previous studies with doped ZrO₂-based catalysts. However, we have also mentioned that this method is not suitable for determining the concentration of surface anion vacancies because bulk oxygen vacancies are also titrated due to high ionic conductivity of such materials. In addition, Zr_{cus} located at corners and edges cannot be titrated by this method. As our samples differ in the size of crystallites, the ratio of Zr_{cus} located at corners and edges to Zr_{cus} located on the flat surface will strongly change from sample to sample. Therefore, to avoid any misinterpreting in the present study, we used a surface-sensitive technique such as TPD of CO, NH₃ or C₃H₆ (Fig. 5d, Supplementary Fig. 14) with these molecules being adsorbed on Zr_{cus}. A correlation between the number of adsorption sites and the size of crystallites or catalyst activity was established (Fig. 5d, Supplementary Fig. 14, Supplementary Fig. 15).

It is difficult to give a precise answer about the critical crystallite size beyond which the activity decreases due to the following reason. ZrO₂ samples with small crystallites are typically prepared at temperatures well below 550°C, where the PDH reaction was investigated. The crystallite size will, however, increase when ZrO₂ will be exposed to 550°C as we have actually observed in our study (Supplementary Table 2). Thus, the main challenge for answering this question properly will be stabilization of small crystallites against sintering.

Comment 4

The zig-zag edges for the smaller crystallites is not everywhere(Fig 3a) and it will be a qualitative interpretation to link this observation to a large number of corner atoms in smaller crystallites.

Reply to comment 4

We agree with the reviewer that Fig. 3a in the original manuscript (Fig. 5a in the emended manuscript) is just a qualitative illustration. Such structures were, however, found in different places upon TEM measurements of ZrO_2 with 9.1 nm crystallites. There were also particles without the zig-zag edges. However, such structures were never observed in amorphous ZrO_2 and ZrO_2 with 43.4 nm crystallites.

The manuscript has been emended on page 12 as follows.

“Zigzag edges were seen in some images of the sample with 9.1 nm crystallites (Fig. 5a) and indicate the presence of a large number of corner atoms. Such surface defects should be coordinatively unsaturated zirconium and/or oxygen ions. They were not observed in the sample with 43.4 nm crystallites. Nearly perfect lattice planes with no corner atoms are typical for the sample with 43.4 nm crystallites (Fig. 5b), while no lattice planes are visible for the XRD-amorphous sample (Fig. 5c).”

Comment 5

The CO TPD/TPR and the conductivity tests for amorphous zirconia needs to be compared with the crystallite zirconia in the characterisation.

Reply to comment 5

As the reviewer suggested, we have performed additional tests with an amorphous ZrO_2 sample and added the obtained results to the emended manuscript (Fig. 5, Supplementary Fig. 13, Supplementary Table 5). This sample showed low conductivity, reducibility and CO adsorption capacity.

Comment 6

Interestingly, propene selectivity also increased with decreasing crystallite size (Figure 1b). What could be the reason for this? Is it possible to have experimental or theoretical insight on this?

Reply to comment 6

This is an important and intriguing question. In general, coke formation in alkane dehydrogenation is very complex and less understood in comparison with propene formation. Deep mechanistic studies are still missing. As established in our previous studies on propane dehydrogenation over non-doped VO_x -based catalysts [ChemCatChem 2015, 7, 1691, Journal of Catalysis 352 (2017) 256], the acidity is not so important for coke formation as the speciation of supported VO_x species. The larger the species, the higher the selectivity to coke was. A probability of interaction between several adsorbed propylene to form heavier hydrocarbons being coke precursor was suggested to increase upon increasing the size of VO_x species where propene molecules adsorb. A similar explanation can be used in the present study when assuming that coke precursors are preferentially formed over acidic Zr_{cus} sites located on the flat surface and not on those located at corners or edges. This statement might be indirectly supported by the new results of our operando UV-vis study. The kind of coke species was suggested to depend on the size of ZrO_2 crystallites (Fig. 2 in the emended manuscript).

The manuscript has been emended on pages 7-9 as follows.

“The use of ZrO_2 composed of small crystallites is also beneficial for propene selectivity, which increased with a decrease in the crystallite size (Fig. 1b). Such positive effect holds over a broad range of propane conversion (Supplementary Fig. 6). An insight into the formation and the kind of carbon deposits was derived from the operando UV-vis analysis upon PDH over three ZrO_2 samples with 9.1, 13.0, or 43.4 nm crystallites. UV-vis spectra expressed as the relative Kubelka-Munk function ($F(R_{rel.})$ in equation (2)) after different times on propane stream are

shown in Fig. 2a-c. $F(R_{rel.})$ increased across the entire range of the UV-vis spectrum with time on propane stream. This increase must have occurred due to the deposition of coke species because the UV-vis spectra of H_2 -treated catalysts differ strongly (Supplementary Fig.7). Absorption bands with the maxima at about 300, 420, 600 and 750 nm can be tentatively identified in the spectra of ZrO_2 under PDH conditions. The latter two signals can be ascribed to polyaromatic graphitic species as previously suggested for PDH over a Cr-containing catalyst²⁵. Temporal changes of the intensity of these bands under PDH conditions follow the same trend thus suggesting that they belong to the same carbon-containing species (Fig.2 d-f). The bands with the maxima in UV range (at 300 and 420 nm) should originate from low-condensed aromatic species, which differ from those also absorbing light above 500 nm. Different carbon-containing species are formed with different rates as seen in Fig. 2d-f showing temporal changes in the intensity of the corresponding bands with rising time of propane stream. When analysing the UV-vis spectra in Fig. 2a-c, it becomes obvious that the relative ratio of the intensity of absorption bands at 300 and 420 nm to that of the bands at higher wave lengths depends on the size of ZrO_2 crystallites. The larger the size, the higher the fraction of polyaromatic graphitic species (absorption bands above 500 nm) is expected. In addition, there is an induction period, before the UV-vis spectra start to change under the PDH conditions (Supplementary Fig.8). Such delay may indicate that some coke precursors (seeds) must be initially formed before coke formation can proceed. Importantly, the duration of this induction period increased with a decrease in the size of ZrO_2 crystallites (Supplementary Fig.8).

On the basis of the above discussion, the effect of the size of ZrO_2 crystallites on coke formation can be explained as follows. From a mechanistic point of view, adsorbed propylene molecules appears to interact with each other to initially form small aromatic structure followed by further oligomerization and condensation to large graphitic structures¹⁵. It is reasonable to suggest that propene molecules adsorbed horizontally to catalyst surface can recombine owing to their spatial location. Such adsorption should be more favourable for Zr sites located on flat ZrO_2 surfaces in comparison to those located on corners or edges. Upon increasing the size of crystallites, the fraction of the former species will decrease and thus will result in a higher rate of carbon deposition. Exactly this trend was observed experimentally. Further experimental and theoretical studies are, however, required to clarify the size effect on coke formation.”

Comment 7

How does crystallite size affects the activation energy of the reaction? How does the Arrhenius factor contribute to the rate of the reaction at smaller crystallite size?

Reply to comment 7

As the reviewer requested, the emended manuscript contains the information about the effect of crystallite size on apparent activation energy of propene formation (Supplementary Fig. 3b). The corresponding Arrhenius plots are shown in Supplementary Fig. 3a. An increase in the activation energy was observed with an increase in the size of crystallites. This result may indicate that the intrinsic dehydrogenation property of Zr_{cus} sites depends on their location, i.e. corners, edges vs. flat surfaces. There is also an alternative explanation. According to our DFT calculations, C_3H_8 dehydrogenation can take place on both regular and defective ZrO_2 surfaces, i.e. in the absence or the presence anion vacancies. The energy for this reaction is, however, higher for the former case than for the latter case. On the basis of these results, the measured overall activation energy will depend on the ratio of defective to non-defective sites and should decrease with rising ratio as the defective surface opens a new energetically more efficient reaction pathway of alkane dehydrogenation.

The manuscript has been emended on pages 5-6 as follows.

“The energy increases from about 140 to 300 kJ·mol⁻¹ with an increase in the size from 9 to 43 nm (Supplementary Fig. 3). This effect is related to the kind of catalytically active sites as will be demonstrated below when taking further catalytic and characterisation studies as well as DFT calculations into account.

In general, it is difficult to determine the Arrhenius factor from the intercept of the dependence of logarithm of the reaction rate on reciprocal temperature (Arrhenius plot) with axis Y. This intercept also includes the concentration of catalytically active sites, which are unfortunately not known.

Comment 8

Will it work with other oxides like Titania? or it is specific to zirconia alone?

Reply to comment 8

Our first attempts with TiO₂ indicate that this material can also catalyse alkane dehydrogenation. The activity is, however, lower than that of ZrO₂. Further studies are in progress to elucidate the potential of TiO₂-based catalysts for this reaction.

Reply to Reviewer 3

Comment 1

Previous works have investigated the effect of crystalline sizes on Metal-Oxygen (M-O) coordination and the C-H activation. The results showed that metal oxides with larger particle sizes have longer M-O bonds, leading to the favorable formation of uncoordinated M sites (J. Catal. 293, 175-185, ACS Catal. 7, 4, 2419-2424), which is contrast to this work.

Reply to comment 1

Thank you for the papers you mentioned. Such difference between the literature data and our results can be due to various reasons. First of all, those previous papers dealt with $\text{CuO}_x/\text{CeO}_2$ and LaFeO_3 , which obviously differ from ZrO_2 from a chemical point of view. Moreover, for the former system, the supporting material CeO_2 also had an effect on the Cu-O bond length. For the LaFeO_3 system, the lowest size of crystallites was 24 or 61 nm for the fresh or spent sample. We established that the reducibility of ZrO_2 strongly increased with a decrease in the size of ZrO_2 crystallites down to 9 nm. This result is in agreement with previous DFT calculations predicting that the energy for the formation of an oxygen vacancy in ZrO_2 lowers for nanoscale particles [ACS Catal. 7, 6493-6513 (2017)].

Comment 2

The bulk ZrO_2 showed decreasing C_3H_6 formation and increasing carbon deposit with time. Due to C_3H_8 , C_3H_6 and H_2 can serve as reductive gases, how about the structural evolutions of such ZrO_2 under operating conditions.

Reply to comment 2

Our previous in situ XRD measurements of ZrO_2 -based materials in a H_2 -containing flow up to 750°C did not reveal any changes in phase composition. The only change we observed in the present study is an increase in the size of crystallites caused by temperature-induced sintering process under propane dehydrogenation conditions.

Comment 3

The authors present the mean crystalline sizes of ZrO_2 by XRD, however the morphologies and sizes are not uniform. Therefore, correlating C_3H_6 formation with crystalline sizes directly may be not correct. Moreover, the authors should present the crystalline sizes with error bars. In addition, what about the XRD and Raman peak shifts when oxygen vacancies were generated?

Reply to comment 3

We agree with the reviewer that the morphology can play a role. We have even written in the Conclusions "Although the concentration of Zr_{cus} can be adjusted in a controlled manner through the size of ZrO_2 crystallites, further improvements are expected when the coordination number of zirconium cations, their location on certain facets and/or the shape of crystallites can also be tuned." Nevertheless, the dependence of the rate of propene formation on the size of crystallites in Fig. 1a was obtained on the basis of catalytic tests with about 40 different monoclinic ZrO_2 samples. This correlation strongly supports the importance of the size of crystallites.

The size of crystallites determined from XRD is an average value among about $1 \cdot 10^{20}$ crystallites located in the sample probe. Therefore, the obtained value is statistically validated. The adjustment of the used diffractometer is weekly checked according to the Si (111) reflection peak at 28.433° . The deviation of the peak position is within the standard deviation (0.004°). To provide a further support, we have measured XRD of five randomly taken probes of ZrO_2 with

9.1 nm, which is one of the lowest values. The size of crystallites was determined with an error of 0.02 nm.

As the number of oxygen vacancies formed upon catalyst treatment in H₂ at 550 °C is very low compared to the number of lattice oxygen ($1 \cdot 10^{16}$ vs. $1 \cdot 10^{22}$ per gram), no changes in the lattice parameters of ZrO₂ were unfortunately observed.

The manuscript has been emended on page 19 as follows.

“The adjustment of the used diffractometer is weekly checked according to the Si (111) reflection peak at 28.433°. The deviation of the peak position is within the standard deviation (0.004°). We have also measured XRD of five randomly taken probes of ZrO₂ with 9.1 nm, which is one of the lowest values. The size of crystallites was determined with an error of 0.02 nm. Thus, the error in determining the size of crystallites is below 1%.”

Comment 4

Previous work (Catal. Sci. Technol., 2017, 7, 4499-4510) has shown that the ZrO₂ with small crystalline sizes present more acidic sites, which bind C₃H₆ strongly and thus lead to further dehydrogenation to form coke. However, why did the smaller ZrO₂ with more acidic sites show lower coke selectivity in this work? Also, smaller ZrO₂ should show less sintering-resistant ability during dehydrogenation and redox cycles, which differs from this work.

Reply to comment 4

Our previous study dealt with ZrO₂ composed of monoclinic and tetragonal phases. At that time, we were not able to prepare pure monoclinic ZrO₂. As a consequence, a direct comparison with the present study is not possible. Moreover, ZrO₂ from that previous study showed higher coke selectivity and concentration of acidic sites in comparison to ZrO₂ promoted with La₂O₃, Y₂O₃ or Sm₂O₃. The promoters are actually of basic character and changed the acidity of ZrO₂ accordingly. It is well known that basic promoters hinder coke formation.

In the present study, we used ZrO₂ mainly composed of the monoclinic phase but differed in the size of crystallites. No promoters affecting material acidity were applied. As determined in our previous studies on propane dehydrogenation over non-doped VO_x-based catalysts [ChemCatChem 2015, 7, 1691, Journal of Catalysis 352 (2017) 256], the acidity is not important for coke formation, while the speciation of supported VO_x species plays a role. The larger the species, the higher the selectivity to coke was. A probability of interaction between several adsorbed propylene to form heavier hydrocarbons being coke precursor was suggested to increase upon increasing the size of VO_x species. A similar explanation can be used in the present study when assuming that coke precursors are preferentially formed over acidic Zrcus sites located on the flat surface and not on those located at corners or edges. The statement might be indirectly supported by the results of our operando UV-vis study. The kind of coke species was found to depend on the size of ZrO₂ crystallites (Fig. 2 in the emended manuscript). The manuscript has been emended on pages 7-9 as follows.

“The use of ZrO₂ composed of small crystallites is also beneficial for propene selectivity, which increased with a decrease in the crystallite size (Fig. 1b). Such positive effect holds over a broad range of propane conversion (Supplementary Fig. 6). An insight into the formation and the kind of carbon deposits was derived from the operando UV-vis analysis upon PDH over three ZrO₂ with 9.1, 13.0, or 43.4 nm crystallites. UV-vis spectra expressed as the relative Kubelka-Munk ($F(R_{rel.})$ in equation (2)) function after different times on propane stream are shown in Fig. 2a-c. $F(R_{rel.})$ increased across the entire range of the UV-vis spectrum with time on propane stream. This increase must have occurred due to the deposition of coke species because the UV-vis spectra of H₂-treated catalysts differ strongly (Supplementary Fig.7). Absorption bands with the

maxima at about 300, 420, 600 and 750 nm can be tentatively identified in the spectra of ZrO_2 under PDH conditions. The latter two signals can be ascribed to polyaromatic graphitic species as previously suggested for PDH over a Cr-containing catalyst²⁵. Temporal changes of the intensity of these bands under PDH conditions follow the same trend thus suggesting that they belong to the same carbon-containing species. The bands with the maxima in UV range (at 300 and 420 nm) should originate from low-condensed aromatic species, which differ from those also absorbing light above 500 nm. Different carbon-containing species are formed with different rates as seen in Fig. 2d-f showing temporal changes of the corresponding bands with rising time of propane stream. When analysing the UV-vis spectra in Fig. 2a-c, it becomes obvious that the relative ratio of the intensity of absorption bands at 300 and 420 nm to that of the bands at higher wave lengths depends on the size of ZrO_2 crystallites. The higher the size, the higher the fraction of polyaromatic graphitic species (absorption bands above 500 nm) is expected. In addition, there is an induction period, before the UV-vis spectra start to change under the PDH conditions (Supplementary Fig.8). Such delay may indicate that some coke precursors (seeds) must be initially formed before coke formation can proceed. Importantly, the duration of this induction period increased with a decrease in the size of ZrO_2 crystallites (Supplementary Fig.8). On the basis of the above discussion, the effect of the size of ZrO_2 crystallites on coke formation can be explained as follows. From a mechanistic point of view, adsorbed propylene molecules appears to interact with each other to initially form small aromatic structure followed by further oligomerization and condensation to large graphitic structures¹⁵. It is reasonable to suggest that propene molecules adsorbed horizontally to catalyst surface can recombine owing to their spatial location. Such adsorption should be more favourable for Zr sites located on flat ZrO_2 surfaces in comparison to those located on corners or edges. Upon increasing the size of crystallites, the fraction of the former species will decrease and thus will result in a higher rate of carbon deposition. Exactly this trend was observed experimentally. Further experimental and theoretical studies are, however, required to clarify the size effect on coke formation.”

We agree with the reviewer that sintering-resistant of crystallites will depend on their initial size when particularly ZrO_2 was calcined at low temperatures to have small crystallites. This is actually what we observed. A new table (Supplementary Table 2) with the relevant data has been added to the supplementary information. The manuscript has also been emended on page 5 as follows.

“It is worth mentioning that for ZrO_2 samples calcined at temperatures below 550°C, we used the size of crystallites determined after performing the PDH reaction (Supplementary Table 2).”

Comment 5

The authors quantified the surface oxygen vacancies using TPD measurement, how about the XPS results?

Reply to comment 5

We have previously tried to use XPS for such quantifying surface oxygen vacancies in ZrO_2 promoted with CrO_x [Chem. Commun., 2016, 52, 8164-8167]. Zr^{4+} was the only state in the oxidized sample (EB = 182.2 eV for $\text{Zr}3d5/2$). The Zr 3d spectrum of the sample reduced in H_2 at 750°C could not be well-resolved as in the oxidized counterpart. A feature at lower binding energies appeared thus indicating the formation of Zr suboxides with an O/Zr ratio lower than 2.

Comment 6

The authors prepared different sizes of ZrO_2 through different methods and at different calcination temperatures, even much lower than the reaction temperatures. Is there any structural changes of the low-temperature-prepared catalysts during reactions?

Reply to comment 6

No changes in phase composition (according to XRD) were observed. However, the size of crystallites slightly increased particularly for samples calcined at lower temperature than 550°C, i.e. the temperature of PDH tests. For such materials, the size of crystallites after the PDH reaction was considered. Supplementary Table 2 shows a list of ZrO₂ catalysts prepared at temperatures below 550°C, the size of crystallites and the specific surface area in their fresh form and after the PDH reaction at 550°C. These new data have been added to the supplementary information.

Reviewers' comments:

Reviewer #1 (Remarks to the Author):

From my previous reviews, no comments have been appropriately considered. The reviewers thought the work to be too preliminary, proposed couple of avenues to be explored. For instance, activity should rather be discussed as a function of surface area rather than crystalline size; catalysis is a surface phenomenon and is typically not related to size unless specific facets would be exposed. This and other comments have been ignored.

The current version has still show severe issues that need to be addressed prior to publications.

The Reviewer proposes to reject the manuscript.

Find below some of the concerns:

- 1) Some terms or statements in the introduction are unjustified and in some cases scientifically inaccurate. For example "Recently, we developed eco-friendly and cost-efficient catalysts on the basis of ZrO₂, which had, however, to be promoted with metal oxides and contain supported Ru or Rh NP." There is no part of this statement which should be considered cost-efficient. Other statements such as "ethane steam cracking and the dehydrogenation of propane or iso-butane are environmentally friendly technologies using shale gas..." shows a clear misunderstanding of the environmental evaluation of processes. These statements are misleading to the general readership of the journal and not justified.
- 2) The authors have also an oversimplified view of particles and metal-oxides as evidenced from statements such as "owing to the well-defined structure of nanoparticles of metals or metal oxides." The reviewer does not understand how nanoparticles can be considered as well-defined (even for mono-dispersed one – not discussed here – one can discuss whether to consider them as mono-dispersed or not).
- 3) With regards to catalytic performance in alkane dehydrogenation, it is difficult to know what numbers are reliable and also which numbers differ from the authors previous work. First, and most concerning, is the nearly identical similarities in rates of propane dehydrogenation in comparison to the authors previous work investigating Ru/Rh-doped ZrO₂ materials which utilized small amount of dopants (0.05-0.005 wt%) (listed in Supplementary Table 4 with references). These small levels of dopants translate to milligrams or fraction of milligrams present in the mass of catalyst used while investigating catalytic activities. To make claims that ZrO₂ outperforms the previously doped materials, supported precious metals, and is purely responsible for hydrocarbon dehydrogenation, ICP-MS indicating absence of small precious metal contaminants and control reactions with no catalyst in the reactor bed must be included.
- 4) Some of the authors previous studies (Catal. Sci. Technol. 2017, 7, 4499) have already described ZrO₂ as a catalyst where oxygen vacancies were attributed to the activity in dehydrogenation. So it is not clear what is the scientific contribution of the experimental work in

the current manuscript. The observation that higher surface areas produce more C₃H₆ g cat-1 is not scientifically surprising (see comments above).

5) Considering the authors emphasis in developing structure activity relationships, the presented mechanistic considerations are ambiguous. With regards to the computational work, it is not evident to the reviewer that the "defect/vacancy pathway" would be more probable than the "Zr-O pathway." The entire energy span of the "defect/vacancy pathway" is 2.28 eV and the "Zr-O pathway" is 2 eV. H₂ elimination from the former also represents a highly endothermic reaction step (1.4 eV). And C-H activation represents the highest barrier for the "Zr-O pathway" at 1.25 eV. Considering these points, it is not unambiguous that the calculations suggest the "defect/vacancy pathway" to be favorable.

6) There are also several additional points regarding structure-activity relationships that are not clear. Based on the authors claims that CH₄ conversions results in 100% product selectivity, it is unclear from their mechanistic considerations how C-C bond formation occurs and why this occurs with such high selectivity. In addition, it is not clear how methane can be involved in dehydrogenation per se because it lacks beta C-H bond and cannot thus involve similar elementary steps. Considering the distinction between heterolytic and homolytic C-H bond activation in propane, the authors do very little in comparing experimental rates of isobutane and propane dehydrogenation to further address these mechanistic considerations (it is surprising that isobutane and propane have similar rates).

7) The selectivities presented are considerably lower when compared to industrial systems and similar catalysts. Additionally, ca. 10% selectivity towards coke in these systems is considerably high and does not address concerns of frequent regeneration associated with industrial systems. This also brings concerns to the viability of using a catalyst which requires a two-cycle regeneration for optimal catalytic activity (oxidation to remove coke and reduction to generate active sites). While the authors do mention that propane can serve as the reductant, it is not clearly addressed how this effects activities. And lastly, the argument that the catalyst is cost-efficient is also debatable considering the significantly higher price of ZrO₂ compared to industrial utilized Al₂O₃ materials.

Reviewer #3 (Remarks to the Author):

The authors have addressed my comments. I recommend the publication of the revised manuscript.

Reply to the comments of Reviewer 1 to NCOMMS-18-08612-T

The scientific comments from the reviewer are highly appreciated by the authors. The manuscript has been revised accordingly. Our detailed reply to the specific comments is given below in blue. **The present and previous (to the original submission) comments of the reviewer are exactly written as in his/her reports.**

General criticism

“From my previous reviews, no comments have been appropriately considered. The reviewers thought the work to be too preliminary, proposed couple of avenues to be explored. For instance, activity should rather be discussed as a function of surface area rather than crystalline size; catalysis is a surface phenomenon and is typically not related to size unless specific facets would be exposed. This and other comments have been ignored.

The current version has still show severe issues that need to be addressed prior to publications. The Reviewer proposes to reject the manuscript.”

Reply to the general criticism

We are surprised by this general statement without explaining why our previous point-by-point reply, the modifications made in the manuscript and in the supplementary information do not appropriately consider the previous concerns. Particularly, the effect of catalyst surface area on the rate of propane dehydrogenation has been thoroughly discussed. An additional figure has been made as the reviewer suggested. As we are quite unsure why “... no comments have been appropriately considered,” we repeat below his/her original comments (**exactly as they have been written in his/her report**) and our original reply.

Previous comments to the original manuscript and our reply start from here. The reply to the new comments is given hereafter.

Comment 1

Low activity compared to Cr or Pt based systems

Reply to comment 1

Figure 2 in the original manuscript (Figure 4 in the emended manuscript) shows the space time yield (STY) of propene formation over our ZrO₂ and the most active, state of the art catalysts including Cr or Pt containing materials. One can clearly see that only two Pt-based catalysts among 17 previously tested catalysts showed higher STY than our catalyst. The latter material is pure ZrO₂ without any metal or metal oxide possessing dehydrogenation activity. STY values between 1.6 and 3.1 kg(C₃H₆)/h/kg(catalyst) achieved in the present study are attractive from an application point of view.

In addition, we revealed the molecular structure of catalytically active sites and provided fundamental principles controlling their concentration. This knowledge opens a possibility to tune activity of ZrO₂-based catalysts and can be used for preparation of other materials with high activity in alkane dehydrogenation.

Comment 2

Low stability (which is the major issue in all PDH catalyst)

Reply to comment 2

We agree with the reviewer that our catalysts deactivate with time on propane stream but such behaviour is also valid for commercial catalysts, as actually the reviewer wrote "... which is the major issue in all PDH catalyst". A commercial Cr-based catalyst operates for about 10 minutes followed by oxidative regeneration. Depending on reaction temperature, ZrO₂ performs more or less stable for up to 20 min. Its activity can be restored after oxidative catalyst treatment at the dehydrogenation temperature as proven in a series of 70 dehydrogenation/regeneration cycles lasted over 13 days on stream. In comparison to commercially applied Pt-based catalysts, ecologically harmful Cl₂ or Cl-containing compounds are not required to regenerate our catalyst.

Comment 3

Lack of discussion of literature precedent that describes very similar finding on the same or related materials

Reply to comment 3

We are quite unsure what the reviewer means. To the best of our knowledge there are only a few papers dealing with alkane dehydrogenation over bulk ZrO₂-based materials without supported Pt or MO_x (M=V, Cr, Ga, Mo) species, which are actively participating in the reaction. Our group was actually first who reported about such novel catalysts. We have written in the introduction of the original manuscript "Recently, we developed eco-friendly and cost-efficient catalysts on the basis of ZrO₂, which had to be promoted with metal oxides and contain tiny amounts of supported Ru or Rh NP to show high activity^{18,19}. Coordinatively unsaturated Zr (Zr_{cus}) and neighbouring lattice oxygen were suggested to participate in PDH²⁰. Yet, the promoter used to purposefully create Zr_{cus} sites either increased or decreased the activity hence proving the limitations of this classical approach for tailored catalyst design. Here, we describe how nanostructuring of ZrO₂ crystallites enables the identification of the nature of active sites and the control of their concentration without the usage of any dopant or supported species."

The present study deals with non-doped ZrO₂ materials, which have been never published before. In comparison to our previous studies, we provide new fundamental insights into the role of Zr_{cus} sites. "A structural model of the active site was established owing to our multidisciplinary approach combining density functional theory (DFT) calculations with a number of complementary experimental methods including catalytic tests. The active site consists of two Zr cations located at an oxygen vacancy, which homolytically break the C-H bond in alkanes. The kind of this site differs from that previously suggested by some of us for doped ZrO₂-based catalysts^{18, 19} and by other researchers for different metal oxides used for PDH²¹⁻²³, where metal cation and neighbouring lattice oxygen were suggested to form the active site. Bare ZrO₂ designed especially to maximize the concentration of the active sites showed industrially relevant performance in comparison with commercial-like catalysts containing CrO_x or Pt species and to other alternative state-of-the-art catalysts."

Comment 4

Activity should be presented not as a function of particle size but per surface area (and not activity per surface area as a function of particle size).

Reply to comment 4

We cannot fully follow the reviewer's criticism. According to her/his suggestion, the rate (probably related to catalyst amount) will simply increase with the surface area. Which important scientific information can be derived from such presentation? How will such dependence help to understand how and if the surface area affects catalyst activity?

Our experimental data were actually reported in form of propene formation rate related to either catalyst amount (Fig. 1a) or specific surface area (Supplementary Fig. 2) The latter form is typically used to take into account the effect of specific surface area of different heterogeneous catalysts on their activity. Regardless of the rate expression form, catalyst activity increases with a decrease in the size of crystallites thus proving that the size but not only the surface area determines the activity.

Nevertheless, as the reviewer suggested we have prepared two additional figures showing the rate of propene formation related to catalyst amount or specific surface area versus the surface area. The latter dependence clearly demonstrates that the surface area is not the only factor affecting catalyst activity. To further support our statement, we have provided an additional figure to the Supplementary information (Supplementary Fig. 2). This figure shows apparent activation energy of propene formation over ZrO_2 differing in the size of crystallites. The energy increases from about 140 to 300 $\text{kJ}\cdot\text{mol}^{-1}$ with an increase in the size from 9 to 43 nm.

The manuscript has been emended on page 5 as follows.

"The obtained relationship between the rate and the size of crystallites is also valid when the rate is calculated with respect to catalyst surface area (Supplementary Fig. 2a). Thus, the PDH activity of ZrO_2 should be governed by the number of the active sites. A simple effect of the crystallite size on the activity can be excluded, as the rate related to catalyst surface area increases with the area (Supplementary Fig. 2b-c). No effect of the area would be visible if this catalyst property determined the activity exclusively. A further experimental support for the importance of the size of crystallites for PDH is the fact that apparent activation energy of propene formation depends on the size. The energy increases from about 140 to 300 $\text{kJ}\cdot\text{mol}^{-1}$ with an increase in the size from 9 to 43 nm (Supplementary Fig. 3). This effect is related to the kind of catalytically active sites as will be demonstrated below when taking further catalytic and characterisation studies as well as DFT calculations into account."

Comment 5

The potential energy surface display very high barrier of energy, which indicate that the reaction should not take place 600°C. In addition, the reaction energy path should be given in Free Energy (which will make things even worth)

Reply to comment 5

As the reviewer suggested, we have calculated the Gibbs free energies and total energies of all relevant states (including transition states) along the reaction coordinate. The corresponding diagram has been added to the Supplementary information (Supplementary Fig. 17). However, it should be noted that both propane and propene do not follow the ideal gas rule and this makes such a pressure and temperature dependent reaction more

complicated than expected on the basis of the ideal gas models. Therefore, the results in Supplementary Fig. 17 are to be used only for qualitative comparison. Nevertheless, the calculations predict that the defect surface is more favourable for PDH than its ideal counterpart.

In addition, we also tested the corrections of dispersion (DFT+D3) and Hubbard term (PBE+ U_{eff}). These new data are now presented in the Supplementary Materials (Supplementary Fig. 22 and Supplementary Fig. 23).

It is not obvious that high energy barrier means that the reaction will not take place.

General statement

Overall the results are too preliminary

Reply to the general statement

This statement is surprising for us because the reviewer did not explain why the results are preliminary when particularly taking into account our multidisciplinary approach combining various complementary experimental techniques with density functional theory calculations. This combination enabled us to conclude, for the first time, what the active sites for alkane dehydrogenation over ZrO_2 are and how their concentration can be tuned. The kind of this site differs from that previously suggested by some of us for doped ZrO_2 -based catalysts and by other researchers for different metal oxides used for propane dehydrogenation, where metal cation and neighbouring lattice oxygen form the active site.

The comments to the revised manuscript and our reply are now presented.

Comment 1

Some terms or statements in the introduction are unjustified and in some cases scientifically inaccurate. For example "Recently, we developed eco-friendly and cost-efficient catalysts on the basis of ZrO_2 , which had, however, to be promoted with metal oxides and contain supported Ru or Rh NP." There is no part of this statement which should be considered cost-efficient. Other statements such as "ethane steam cracking and the dehydrogenation of propane or iso-butane are environmentally friendly technologies using shale gas..." shows a clear misunderstanding of the environmental evaluation of processes. These statements are misleading to the general readership of the journal and not justified.

Reply to comment 1

Concerning the reviewer's worry related to costs of ZrO_2 -based catalysts, please, consider the following information. The cost statement has been made in our previous studies [Refs. 18-20] dealt with Me/YZO_x or $Me/LaZrO_x$ ($Me=$ Ru, Rh or Cu) materials. We agree with the reviewer that ZrO_2 is more expensive than Al_2O_3 used as support in commercial Pt-containing catalysts. The difference in price is between 3-5 times. However, Pt is much more expensive than Al_2O_3 , i.e. about 30,000 times. As a consequence, Pt determines the cost of the commercial catalysts although they contain only about 0.5 wt% Pt. Even such low amount contributes to the cost about 150 times higher than the cost for Al_2O_3 . The highest amount of metal in our previous catalysts was 10 times lower than Pt, i.e. 0.05 wt%.

Moreover, Ru or Cu are about 20 or 6,000 times cheaper than Pt. The catalysts presented in the present manuscript are bare ZrO₂ without any additional metal or metal oxide. Therefore, they are even less expensive than those used in our previous studies. We think that the statement about cost efficiency of our catalysts is justified.

Our text about environmental issues was probably not precise and, therefore, provoked the criticism of the reviewer. To avoid any misunderstanding, we have modified this part in the manuscript as follows. "In comparison with oil-based cracking technologies, which provide a major part of C₂-C₄ olefins for the chemical industry, ethane steam cracking and the dehydrogenation of propane or iso-butane are more environmentally friendly processes because they are based on natural/shale gas containing significantly less impurities than the oil-based feedstock." Table 10 in the handbook of heterogeneous catalysis (2008, section 13.5.6 page 2767) shows air pollutants and their estimated uncontrolled emissions in gram per ton processed feed from fluid catalytic cracking. Such pollutants are CO, SO_x, NO_x, NH₃, aldehydes, particulates as well as organic and inorganic hazardous air pollutants. Since propane, ethane or iso-butane feed components contain significantly less impurities than oil-based feedstock for FCC or steam cracking the PDH technology can be considered to be more environmentally friendly than the cracking processes.

Comment 2

The authors have also an oversimplified view of particles and metal-oxides as evidenced from statements such as "owing to the well-defined structure of nanoparticles of metals or metal oxides." The reviewer does not understand how nanoparticles can be considered as well-defined (even for mono-dispersed one – not discussed here – one can discuss whether to consider them as mono-dispersed or not).

Reply to comment 2

We apologize if we were not precise enough when writing "... well-defined structure...". We meant that nanoparticles can be prepared with a certain narrow size distribution, shape and composition.

The manuscript has been emended on page 3 as follows.

"For example, owing to the well-defined structure (size and/or shape) and composition of nanoparticles of metals or metal oxides, they are successfully used for the development of supported catalysts with controlled properties³⁻⁸."

Comment 3

With regards to catalytic performance in alkane dehydrogenation, it is difficult to know what numbers are reliable and also which numbers differ from the authors previous work. First, and most concerning, is the nearly identical similarities in rates of propane dehydrogenation in comparison to the authors previous work investigating Ru/Rh-doped ZrO₂ materials which utilized small amount of dopants (0.05-0.005 wt%) (listed in Supplementary Table 4 with references). These small levels of dopants translate to milligrams or fraction of milligrams present in the mass of catalyst used while investigating catalytic activities. To make claims that ZrO₂ outperforms the previously doped materials, supported precious metals, and is purely responsible for hydrocarbon dehydrogenation, ICP-MS indicating absence of small precious metal contaminants and control reactions with no catalyst in the reactor bed must be included.

Reply to comment 3

One of the purposes of our study was to check if bare ZrO_2 can show high PDH activity in comparison with previously tested ZrO_2 promoted with metal oxides and additionally contained supported Ru or Rh nanoparticles. The latter catalysts serve as benchmark materials. As proven in our present study, bare ZrO_2 can become very active through tuning the nanostructure of crystallites.

All ZrO_2 materials used in the present study have **not** been promoted by any metal oxide or metal. According to the information of the suppliers of used chemicals, trace metal analysis revealed below 150 ppm impurities. Our ICP-OEC failed to determine precisely less than 0.001 wt% supported metals.

Blank propane conversion has been determined in our earlier studies and is about 0.20 or 1.6 % at 550 or 625 °C [J. Catal. 348, 282-290 (2017)], with these values being significantly lower than with catalyst present under identical reaction conditions. Moreover, Figure 1 in the present study shows that the differently prepared ZrO_2 materials differ in their activity by a factor of up to 70. If any non-catalytic reaction played a significant role in our tests, we would not be able to see such strong differences between the catalysts.

The manuscript has been emended on page 5 as follows.

“As seen in Fig. 1a, the rate of propene formation in PDH decreased by a factor of up to 70 with an increase in the size from 7 to 45 nm. If non-catalytic propane dehydrogenation played a significant role, we would not be able to see such strong differences between the catalysts in terms of the rate of propane formation. Separate tests¹⁹ without catalysts proved that non-catalytic propane conversion is not relevant under the reaction conditions applied.”

Comment 4

Some of the authors previous studies (Catal. Sci. Technol. 2017, 7, 4499) have already described ZrO_2 as a catalyst where oxygen vacancies were attributed to the activity in dehydrogenation. So it is not clear what is the scientific contribution of the experimental work in the current manuscript. The observation that higher surface areas produce more C_3H_6 g cat^{-1} is not scientifically surprising (see comments above).

Reply to comment 4

The above previous study has focused on the effect of metal oxide promoter on the activity and selectivity of ZrO_2 -based catalyst. However, this approach has not been proven to be suitable for tailored design and preparation of such catalysts. We have written in the Introduction of the present manuscript “Yet, the promoter used to purposefully create Zr_{cus} sites either increased or decreased the activity hence proving the limitations of this classical approach for tailored catalyst design. Here, we describe how nanostructuring of ZrO_2 crystallites enables the identification of the nature of active sites for efficient C-H bond activation and the control of their concentration without the usage of any dopant or supported species.”

Moreover, owing to sophisticated DFT calculations, the present study concludes that two Zr cations located at an oxygen vacancy form the active site for homolytically breaking the C-H bond in alkanes. Our previous studies with ZrO_2 -based catalysts and studies of other researchers with different metal oxides suggest that lattice oxygen and neighbouring metal cation represent the active site.

Concerning the reviewer's statement about the role of catalyst surface area, we kindly ask, for a second time, to carefully read our reply to his/her original comment 4 (see pages 2-3) and the accordingly emended original manuscript. Supplementary Fig. 2c unambiguously

shows that the surface area is not the only factor affecting catalyst activity. The activity is mainly affected by the size of ZrO₂ crystallites. If the surface area determined the activity, ZrO₂_24 and ZrO₂_26 (amorphous materials) with the highest area of 246.6 and 267.3 m²·g⁻¹ respectively (Supplementary Table 1) would show the highest rate of propene formation related to catalyst amount. As seen in Figure 1a and Supplementary Figure 2 (a-c), they, however, showed the lowest activity.

Further minor modifications have been made on pages 5-6 as follows.

“Moreover, although amorphous ZrO₂ samples (ZrO₂_24 and ZrO₂_26 in Supplementary Table 1) possess the highest surface area, they showed the lowest activity. The PDH activity of ZrO₂ is governed by the number of the active sites, which depends on the size of crystallites. A further experimental support for the importance of the latter catalyst property for PDH is the fact that apparent activation energy of propene formation depends on the size.”

Comment 5

Considering the authors emphasis in developing structure activity relationships, the presented mechanistic considerations are ambiguous. With regards to the computational work, it is not evident to the reviewer that the "defect/vacancy pathway" would be more probable than the "Zr-O pathway." The entire energy span of the "defect/vacancy pathway" is 2.28 eV and the "Zr-O pathway" is 2 eV. H₂ elimination from the former also represents a highly endothermic reaction step (1.4 eV). And C-H activation represents the highest barrier for the "Zr-O pathway" at 1.25 eV. Considering these points, it is not unambiguous that the calculations suggest the "defect/vacancy pathway" to be favorable.

Reply to comment 5

Probably there is a misunderstanding about the entire potential energy surfaces over the d-ZrO₂($\bar{1}11$) and s-ZrO₂($\bar{1}11$) facets in Figure 5.

To understand the thermodynamics, the initial and final states should be considered, while the apparent barrier between the initial state and the highest point should be used to differentiate the kinetics. The reviewer appeared to use the lowest and the highest states.

On each facet, the PDH reaction can be divided into two parts, i.e. propene formation and H₂ formation. The formation of propene over the s-ZrO₂($\bar{1}11$) facet has a barrier of 1.68 eV and is endothermic by 1.40 eV. The formation of H₂ after propene desorption has a barrier of 0.47 eV and is slightly exothermic by 0.29 eV. The apparent barrier is 2.00 eV

In comparison with the s-ZrO₂($\bar{1}11$), propene formation over the d-ZrO₂($\bar{1}11$) facet has a barrier of 0.16 eV only and is exothermic by 0.62 eV, while H₂ formation is uphill and endothermic by 1.40 eV. The apparent barrier of the whole process is 1.24 eV.

Thus, the apparent barrier over the d-ZrO₂($\bar{1}11$) facet is 0.76 eV lower than that over the s-ZrO₂($\bar{1}11$) facet (1.24 vs. 2.00 eV) and this suggests that the entire propane dehydrogenation is kinetically more favoured on the d-ZrO₂($\bar{1}11$) facet than on the s-ZrO₂($\bar{1}11$) facet.

The above statement is also supported by the Gibbs free energy profiles given in Supplementary Fig. 17. This figure shows that the apparent free energy barrier for propane dehydrogenation over the d-ZrO₂($\bar{1}11$) facet is 1.66 eV lower than that over the s-ZrO₂($\bar{1}11$) facet (2.52 vs. 4.18 eV) indicating that the dehydrogenation reaction is kinetically more favoured over the d-ZrO₂($\bar{1}11$) facet than on the s-ZrO₂($\bar{1}11$) facet.

The manuscript has been emended on page 16 as follows.

“This figure shows that the apparent free energy barrier for propane dehydrogenation over the d-ZrO₂($\bar{1}11$) facet is 1.66 eV lower than that over the s-ZrO₂($\bar{1}11$) facet (2.52 vs. 4.18 eV) indicating that the dehydrogenation reaction is kinetically more favoured over the d-ZrO₂($\bar{1}11$) facet than on the s-ZrO₂($\bar{1}11$) facet.”

Comment 6

There are also several additional points regarding structure-activity relationships that are not clear. Based on the authors claims that CH₄ conversions results in 100% product selectivity, it is unclear from their mechanistic considerations how C-C bond formation occurs and why this occurs with such high selectivity. In addition, it is not clear how methane can be involved in dehydrogenation per se because it lacks beta C-H bond and cannot thus involve similar elementary steps. Considering the distinction between heterolytic and homolytic C-H bond activation in propane, the authors do very little in comparing experimental rates of isobutane and propane dehydrogenation to further address these mechanistic considerations (it is surprising that isobutane and propane have similar rates).

Reply to comment 6

The purpose of CH₄ conversion tests was to demonstrate that ZrO₂ can also break the C-H bond in methane, which is the most inert alkane. No in depth mechanistic studies have been performed in the present study. Nevertheless, on the basis of previous studies on the oxidative coupling of methane [Handbook of Heterogeneous Catalysis, 2008, 6, 3010.] or non-oxidative methane conversion to ethylene [Science, 2014, 344, 616], CH₃ radical is suggested to be formed upon breaking the C-H bond in CH₄. Two such radicals recombine to C₂H₆ followed by dehydrogenation of the latter to C₂H₄. As seen in Figure 3a in the manuscript, ZrO₂ is able to catalyse C₂H₆ dehydrogenation to C₂H₄.

Concerning the high selectivity to C₂H₄, we did not claim that we can get 100% selectivity but mentioned that C₂H₄ was the only detected product. CH₄ conversion was only 0.015 %. Due to limitations of our GC analysis to properly separate traces of C₂H₆ from CH₄ in a large excess of the latter, it was very difficult to precisely analyse C₂H₆. Therefore, we did not report the corresponding data.

The manuscript has been emended on page 10 as follows.

“Ethylene was the only product observed. It is, however, worth mentioning that due to the low CH₄ conversion of only 0.015%, it was difficult to precisely conclude if other products were also formed. Therefore, we do not discuss product selectivity in CH₄ conversion tests. On the basis of previous studies on the oxidative coupling of methane²⁶ and non-oxidative methane conversion to ethylene²⁷, CH₃ radical is suggested to be formed through breaking the C-H bond in CH₄. Two such radicals recombine to C₂H₆ followed by its dehydrogenation to C₂H₄. ZrO₂ is actually able to catalyse the latter reaction (Fig. 3a).”

Comment 7

The selectivities presented are considerably lower when compared to industrial systems and similar catalysts. Additionally, ca. 10% selectivity towards coke in these systems is considerably high and does not address concerns of frequent regeneration associated with industrial systems. This also brings concerns to the viability of using a catalyst which requires a two-cycle regeneration for optimal catalytic activity (oxidation to remove coke and reduction to generate active sites). While the authors do mention that propane can serve as the reductant, it is not clearly addressed how this effects activities. And lastly, the argument that

the catalyst is cost-efficient is also debatable considering the significantly higher price of ZrO_2 compared to industrial utilized Al_2O_3 materials.

Reply to comment 7

We agree that the selectivity to propene over ZrO_2 is lower in comparison to the commercially applied Pt- or CrO_x -containing catalysts, i.e. 83-85% vs. 88-91% [K. J. Caspary, H. Gehrke, M. Heinritz-Adrian, M. Schwefer, in Handbook of Heterogeneous Catalysis, G. Ertl, H. Knözinger, F. Schüth, J. Weitkamp, Eds. (Wiley-VCH Verlag GmbH & Co. KGaA, Weinheim, 2008), vol. 7, chap. 14.6, pp. 3206-3228.]. The difference is however not too large. The reviewer's concern about "the viability of using a catalyst which requires a two-cycle regeneration for optimal catalytic activity" is really surprising for us when particularly taking into account that CrO_x -based catalysts used in the CATOFIN process operate for about 10-30 minutes followed by oxidative regeneration. Heat released upon combustion of coke deposits is used for the endothermic PDH reaction.

Concerning cost-efficiency of ZrO_2 -based catalysts, please, see our thorough reply to comment 1. In addition, we are very surprised that catalyst cost is a criterion to decline publication of fundamental study results.

The statement that propane can reduce ZrO_2 simply means that Zr_{cus} sites will be formed in situ without additional reductive catalyst treatment. Like H_2 or CO , C_3H_8 is a reducing agent and is able to remove lattice oxygen from the lattice of ZrO_2 .

Reply to the comments of Reviewer 3 to NCOMMS-18-08612-T

Statement

The authors have addressed my comments. I recommend the publication of the revised manuscript.

We thank the reviewer for his/her positive evaluation of our research.